# Movement errors during skilled motor performance engage distinct prediction error mechanisms

Ella Gabitov [1,2 ✉], Ovidiu Lungu[1,2], Geneviève Albouy[3] & Julien Doyon [1,2 ✉]

The brain detects deviations from intended behaviors by estimating the mismatch between predicted and actual outcomes. Axiomatic to these computations are salience and valence prediction error signals, which alert the brain to the occurrence and value of unexpected events. Despite the theoretical assertion of these prediction error signals, it is unknown whether and how brain mechanisms underlying their computations support error processing during skilled motor behavior. Here we demonstrate, with functional magnetic resonance imaging, that internal detection, i.e., without externally-provided feedback, of self-generated movement errors evokes instantaneous activity increases within the salience network and delayed lingering decreases within the nucleus accumbens – a key structure in the reward valuation pathway. A widespread suppression within the sensorimotor network was also observed. Our findings suggest that neural computations of salience and valence prediction errors during skilled motor behaviors operate on different time-scales and, therefore, may contribute differentially to immediate and longer-term adaptive processes.

[1] McConnell Brain Imaging Center, Montreal Neurological Institute, Montreal, QC H3A 2B4, Canada. [2] Department of Neurology and Neurosurgery, Montreal Neurological Institute, McGill University, Montreal, QC H3A 2B4, Canada. [3] Movement Control and Neuroplasticity Research Group, Department of Movement Sciences, KU Leuven, 3000 Leuven, Belgium. ✉email: gabitovella@gmail.com; julien.doyon@mcgill.ca

Grabbing a cup of coffee, typing on a keyboard, or driving a car—either of these and many other everyday activities typically involve a sequence of skilled motor actions that are executed precisely, effortlessly, and fast. However, even highly skilled human behavior is susceptible to occasional errors. Although errors during skilled motor performance are rare, they may lead to serious consequences. Yet, little is known about neural networks underlying error processing in such scenarios.

An action error is known to occur when the internal prediction about the consequences of an intended action do not match the actual outcome of that action[1]. Such prediction error signals (PES) not only provide detailed information about the motor command that caused the error, but also induce more generic estimations indicating the degree of salience and/or valence of the mismatch. The salience PES are critical for prompt error detection. They are thought to initiate a cascade of automatic processes[2] that aim at rapid interruption of ongoing behavior, and/or shift of attention to the source of the unpredicted outcome[3]. As such, these alerting signals are derived promptly with minimal neural computations and are typically deemed to be both non-specific and unsigned, i.e., without indicating whether the actual outcome is better or worse than predicted[4]. The valence PES, commonly referred to as reward PES[5], also lack specificity but, unlike salience alert signals, are signed: that is, they are positive when the action outcome is better than predicted (e.g., greater reward, more enjoyable experience, or better performance than expected), and negative when the outcome is worse. By retroactive assignment of credit or blame to actions that led to success or failure, respectively, valence PES may constitute a crucial cue to induce learning[6].

The neural correlates of salience PES have been previously studied building on a seminal finding of error-related negativity—a component of the event-related brain potential. This component reliably indexes error detection and its origin has been attributed to functions of the anterior cingulate cortex (ACC)[7]. A mounting body of evidence from human neuroimaging studies also suggests that the dorsal part of this region (dACC), along with the closely affiliated areas within the anterior insula (aIns), responds to varied forms of salience, including errors[8–14], as well as reward, pain, surprise, and other unexpected events[15–17]. More recently, aaugmented activity within the salience network associated with the occurrence of movement errors was also reported[18]. Furthermore, it has been suggested that the dACC may generate salience PES in situations with increased error likelihood[4,19] and may also accommodate proactive mechanisms of error detection during highly skilled piano performance[20].

The neural correlates of valence PES have been initially demonstrated in animal models using reinforcement learning, during which improved performance maximizes the received reward[5]. It has been posited that dopamine neurons in the midbrain ventral tegmental area (VTA) may be implicated in generation of these PES. The dopamine neurons fire in response to unpredicted reward and pause in response to unexpected omission of reward. Similarly, signed reward-related PES have also been detected in the human brain using functional magnetic resonance imaging (fMRI). Modulation of neural activity associated with reward has indeed been identified in the dopamine target areas, including the ventral striatum, particularly in the nucleus accumbens (NAc)[21,22]—a primary efferent target of the VTA neurons within the mesolimbic pathway[23]. Under certain conditions, however, this dopamine-innervated striatal structure may be involved in the processing of salient events without any reward, feedback, or motivational value[24] and may also promote action initiation[25].

Despite the theoretical assertion of salience and valence PES in the brain, it is unknown whether and how error processing during skilled motor performance relies on neural substrates underlying their computations. Here, we addressed this knowledge gap by investigating behavioral and neurophysiological changes related to errors committed during a self-guided motor sequence task[26], using fMRI. Participants were asked to memorize a short five-element sequence and to tap it repeatedly on a keypad using their nondominant (left) hand[27]. The sequence was reproduced in a continuous and self-paced manner without relying on any external cue or input. These conditions closely resemble situations in everyday life, when errors constitute deviations from the sequential skilled motor movements pre-planned in advance. It is in contrast to errors committed during speeded reaction-time paradigms[28], during which the unpredicted order of stimuli requires exogenous attention and virtually eliminates the possibility to plan motor responses in advance. In addition, in the current study, no feedback nor incentive, in any form, were given to participants. The feedback-free mode of performance not only rules out the possibility of attributing changes associated with errors to the external cue, but also implies that error detection and subsequent performance recovery would be realized via internally driven prediction error mechanisms.

At the behavioral level, we hypothesized that slowing in performance would constitute a behavioral signature of error detection[29]. We also expected that error instances would be characterized by a speed-accuracy trade-off[30]—a phenomenon that has been consistently reported during speeded reaction-time tasks[31]. Such a trade-off suggests that faster performance increases the probability of errors and, therefore, should precede error commission. It also entails that reinstatement of accurate performance after errors would compromise the speed leading to post-error slowing.

At the neurobiological level, we aimed to elucidate (1) whether error processing during skilled motor performance is associated with changes in brain regions implemented in computations of salience and valence PES, and if so, (2) how these changes evolve throughout different phases of error processing. Given the unimanual nature of the task, we also assessed whether interruption of inadequate performance during errors is implemented via global (i.e., widespread and bilateral) or selective (i.e., effector-specific and lateralized) suppression mechanism[32].

We made two key assumptions. First, due to highly accurate performance during the task employed in the current study[27,33], errors would constitute unexpected events triggering unsigned salience PES. Concurrently, this sudden violation of predicted success in a form of errors would also evoke negative valence PES. Second, brain regions previously identified as implicated in PES and those forming the motor-task-related network[34] would exhibit dissociative dynamics during errors. This dissociation would minimize the possibility that error-related changes within the salience network could be attributed to task maintenance—an inherent problem in studies using cognitive and attention-demanding tasks[35]. The dissociation between the error- and task-related networks would be robust, allowing the detection of a reliable error signature even when the number of errors is low[36,37]. We not only reproduced our results for a subgroup of participants, but also conducted a replication study using data acquired during another separate session.

## Results

**Sequences and errors.** The main units of interest in our analysis are trials, i.e., sequences and errors, comprising several keypresses (Fig. 1a). Such an approach suits better the low temporal resolution of the fMRI signal allowing us to assess neural dynamics across trials in a space-resolved manner. To get insights into the

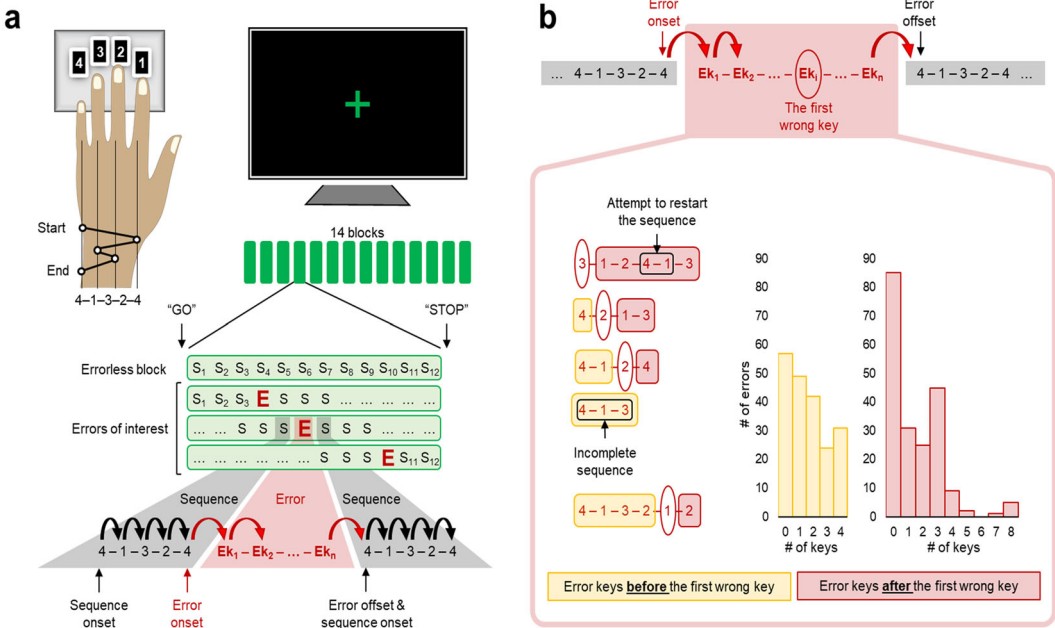

**Fig. 1 Study design and errors of interest. a** Participants were scanned while tapping a five-element sequence (S: 4–1–3–2–4) on a keypad using their left hand. The session consisted of 14 successive performance blocks with 60 keypresses, equivalent to 12 repetitions of the correctly performed and completed sequence (i.e., correct trials; $S_1$, $S_2$,..., $S_{12}$), each block being separated by 15-s periods of rest. During performance blocks, participants were asked to look at a fixation cross and to tap the sequence repeatedly as fast and accurately as possible. In case of errors, they were instructed to continue with the task as smoothly as possible from the beginning of the sequence. A change in the color of the fixation cross from red to green and from green to red indicated the beginning (GO) and the end (STOP) of each performance block, respectively. Only isolated errors preceded and followed by at least three correct trials in a row were analyzed; the last error within the block followed by at least two correct trials in a row was also considered. **b** All wrong keypresses that violated the predetermined order (i.e., sequence) or incomplete sequences with missing keys, between two correct trials, were considered as a single error. Thus, the first wrong keypress that violated the sequential order (marked with red oval) can be the first, second, third, fourth, or fifth error key. S—correct trial, i.e., one correctly performed and completed sequence, E—erroneous trial, i.e., an error, $Ek_1 ... Ek_n$—keys within the error, $Ek_i$—the first wrong key within the error, black rounded arrows—transitions within the sequence, red rounded arrows—error transitions.

types of errors and changes in performance during errors, behavioral data were also analyzed at the level of single keypresses.

Five consecutive keypresses that followed the predetermined order (i.e., sequence) are considered as one correct trial. Any other combination of consecutive keys (or a single key), including instances of incomplete sequence, between two correct trials constitutes an error (Fig. 1b). Thus, the first wrong keypress that violated the predetermined order of the sequence could occur after up to four keys pressed correctly.

**Errors of interest**. We aimed at characterizing changes throughout all phases of error processing, including error commission and the subsequent performance recovery. Therefore, only isolated errors surrounded by at least three correct trials in a row, hence forming periods with seven consecutive trials, were analyzed (Fig. 1a). If the last error within the block was followed by only two consecutive correct trials, that error was also considered. Such an approach allowed us to minimize the overlap between the pre- and post-error intervals, and to reduce the effects of transient nonspecific activity bursts at task initiation and termination[38]. To avoid analyzing extremely short or recurring errors, errors that lasted <0.5 s or comprised >20 keys (i.e., with the number of keys exceeding four correct trials) were excluded.

Examples of possible errors and their distribution with respect to the first wrong keypress are shown in Fig. 1b. Errors (203 errors in total) that met our selection criteria were detected in 49 (out of 54) individuals (4.14 ± 0.30 errors, mean ± SEM; Fig. 2a).

Errors' distribution considering their length, duration, and type is shown in Fig. 2b, c.

**Control task condition**. To determine the specificity of error-related changes, periods without errors were used as a control task condition. Like periods with errors, these periods comprised seven (or six if the period was at the very end of the block) consecutive trials. To account for changes in performance and in neural activity associated with fatigue[39], time-on-task[35], and the position of the trial within blocks (see fMRI results below), periods with and without errors were pseudo-randomly matched by their within-block position ("Methods" section). This position was determined by the fourth trial, which during periods with and without errors constituted an error, and so-called position-matched sequence, respectively. Potential differences in the statistical power between the two conditions (i.e., periods with and without errors), were minimized by including the same number of periods in each condition.

**Distribution of errors and position-matched sequences**. The distribution of errors and position-matched sequences within and across blocks is shown in Fig. 2d. We conducted a detailed analysis to test for heterogeneity of trial distribution within and across blocks. None of the results are statistically significant (effect of within-block position: $F(6, 318) = 1.10$, $p = 0.36$; effect of block: $F(13, 624) = 1.326$, $p = 0.193$; block by trial type interaction: $F(8.95, 429.76) = 1.307$, $p = 0.231$), suggesting non-heterogeneous and comparable distribution of periods included in each condition (i.e., periods with and without errors). The time spent on task from the beginning of the block until the trial onset

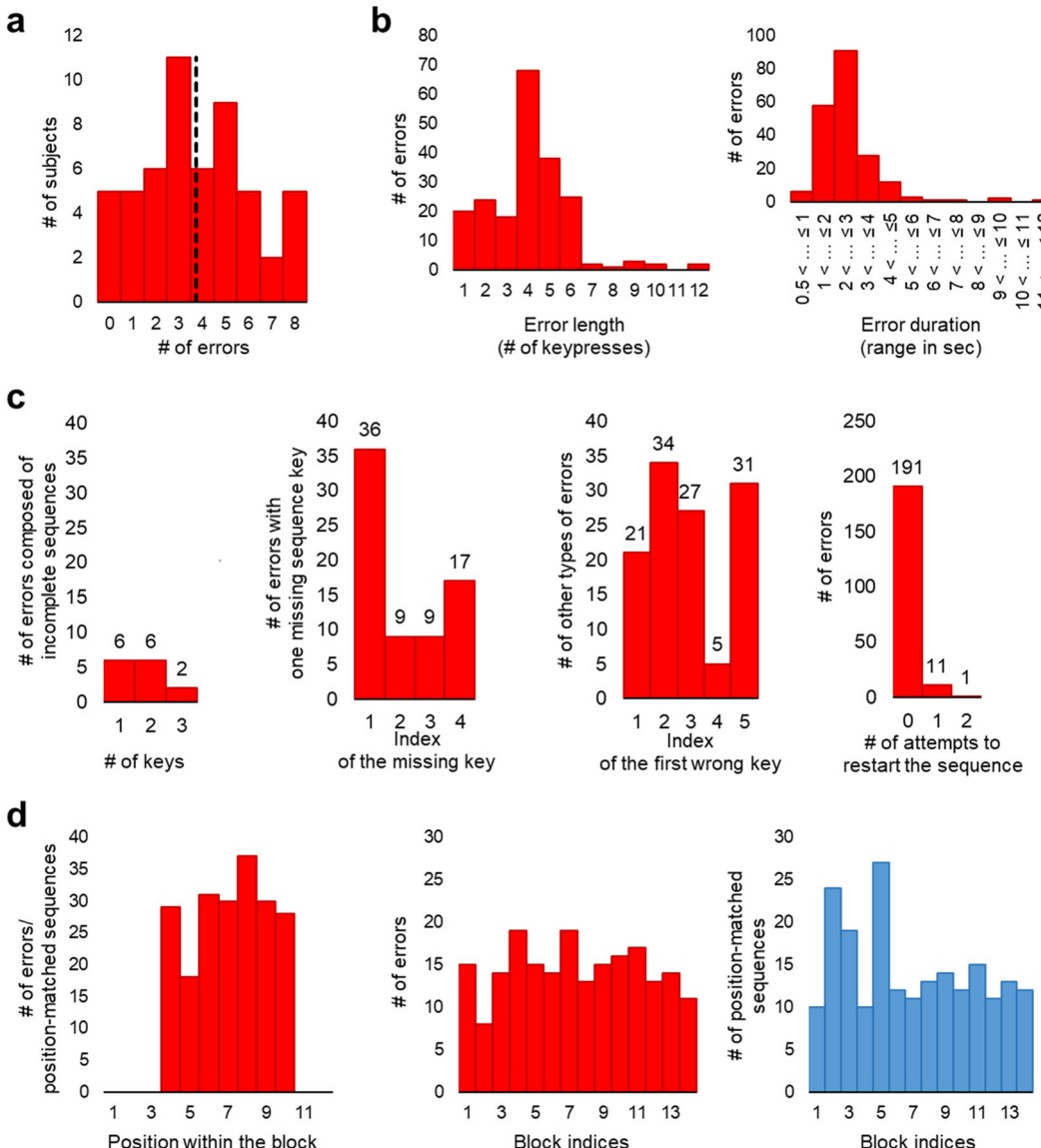

**Fig. 2 Sample summary for errors of interest and position-matched sequences. a** The distribution of number of errors across participants. The vertical dashed line represents the group mean of the number of errors. **b** The number of errors plotted against their length (i.e., the number of keypresses within the error) and duration (i.e., the total time spent on the error starting from its onset until its offset). The vast majority of errors comprise between 1 and 6 keypresses, and last <6 s. **c** The distribution of errors by different types and the number of attempts to reinitiate correct task performance. Here, incomplete sequences are instances of correctly initiated, but unfinished sequence with up to three correctly pressed keys. Unfinished sequences with four correct keys are labeled as errors with one missing key. An attempt to restart the sequence is determined by the first sequence transition, i.e., the key combination of 4–1. **d** The distribution of errors/position-matched sequences within and across performance blocks. Note, that errors and position-matched sequences have the same distribution within blocks.

is also comparable between the two conditions ($t(48) = 1.63$, $p = 0.110$).

**Behavioral results**. Transition duration, i.e., a time interval between two consecutive keypresses, was used as a measure for speed. To estimate speed during correct trials, the mean of four within-sequence transitions was calculated, whereas during errors, transitions to the first and from the last error key were also considered (Fig. 1a)—below, these transitions are referred to as error onset and offset, respectively. To account for changes in performance across blocks, transition durations were converted into percentage relative to the mean transition duration within all correct trials on a block-by-block basis. These values were used to estimate the magnitude of slowing and to assess error-specific

changes in performance. All measures were first calculated at the individual level and then introduced to the group-level analysis.

Mean transition duration and the magnitude of slowing immediately before, during and after errors are shown in Fig. 3a, b. In line with our hypothesis, the tapping speed during errors dramatically drop down ($46.99 \pm 5.38\%$, mean ± SEM) and is significantly slower than during correct trials ($t(48) = 8.13$, $p < 0.001$). We also expected to find evidence for the speed-accuracy trade-off immediately before and after errors in the form of faster and slower tapping speed, respectively. However, a significant deviation from the mean transition duration of correct trials is observed only during the post-error ($t(48) = 3.62$, $p = 0.001$), but not during the pre-error performance ($t(48) = -0.22$, $p = 0.824$), hence providing evidence for the speed-accuracy trade-off only

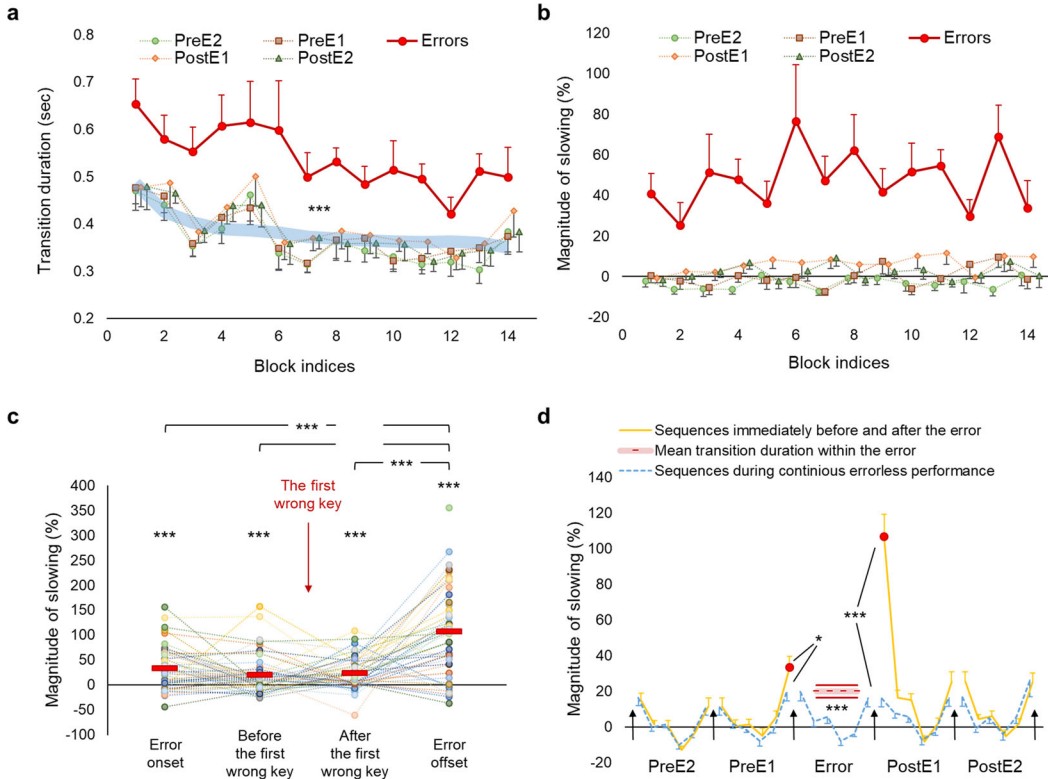

**Fig. 3 Behavioral results. a, b** Mean time to complete a single transition (i.e., transition duration) and the magnitude of slowing during errors (red markers), and within two adjacent sequences immediately before and after the error (colored markers) are plotted for each performance block. The wide light-blue line marks mean transition durations of all correct trials in each block. Data points represent group means for each block. **c** The magnitude of slowing at the error onset, within the error—mean values for transitions before and after the first wrong keypress are shown separately—and at the error offset. Data points represent mean values calculated across errors for each individual. Red markers indicate group means. **d** The magnitude of slowing during periods with and without errors. Orange/blue lines connect mean values for transitions within and between correct trials during these periods. Black arrows indicate transitions between trials. Red circles represent mean values for transitions at the error onset/offset. Dashed and continuous red lines represent the mean magnitude of slowing and standard error of the mean (SEM) within the error, respectively. Zero values of the magnitude of slowing (i.e., the reference line marked by the x-axis) represent the mean transition duration within all correct trials; these values were initially calculated separately for each block, and then used to estimate the magnitude of slowing on a block-by-block basis. PreE2 and PreE1—two consecutive sequences immediately before the error, PostE1 and PostE2—two consecutive sequences immediately after the error. * and *** - significant results at 0.05 and 0.001 level, respectively. Error bars represent SEM.

after errors. Such a trade-off in the form of significant post-error slowing ($6.44 \pm 1.35\%$, mean ± SEM) is transient and rapidly disappears by the second post-error sequence ($t(48) = 1.09$, $p = 0.28$).

To get insights into how performance speed changes during different error phases, we assessed the magnitude of slowing separately at the error onset, within the error, and at the error offset (Fig. 3c; changes in tapping speed during errors grouped by their length are shown in Supplementary Fig. 1). Transitions within the error were further divided into transitions before and after the first wrong keypress (Fig. 1b; in some errors, no keys were pressed before and/or after the first wrong key). A significant drop in performance speed is evident throughout all phases of errors even before the predetermined order of the keypresses is actually violated (i.e., before the first wrong key is pressed; $t(41) > 3.675$, $p < 0.001$; the magnitude of slowing >22.672%). The strongest effect is observed at the error offset ($t(48) = 8.558$, $p < 0.001$ with $106.774 \pm 12.476\%$ of slowing, mean ± SEM), which is also the onset of the subsequent correct trial (Fig. 1a). Thus, the greatest drop in speed coincides with the successful reinitiation of the task. The degree of this post-error slowing surpasses the degree of the reduced speed at the block initiation phase (Supplementary Note 1 and Supplementary

Fig. 2) and, therefore, cannot be fully explained by the need to reinitiate the task per se.

Finally, we also compared the magnitude of slowing during distinct error phases with the corresponding phases of position-matched sequences (i.e., control task condition) and reaffirmed that slowing associated with error commission is, indeed, specific to errors and evident throughout all error phases (Fig. 3d).

**fMRI analysis**. To determine neural substrates mediating error processing, functional data were analyzed using a mixed block/event-related design[40]. Performance periods were modeled as blocks/epochs, whereas trials of interest (i.e., errors and sequences) were modeled as events. To minimize the effects of various trial durations, trial-related changes in the blood-oxygen-level-dependent (BOLD) signal were estimated, using a stick function with zero duration (Methods).

Before analyzing error-related changes, we assessed neural signals captured by the stick function during errorless blocks (Supplementary Fig. 3). These signals fluctuate robustly following a consistent pattern within blocks with no significant differences between blocks ($F(11, 8162) = 72.79$, $p < 0.001$; $F(13, 742) = 1.48$, $p = 0.12$; $F(143, 8162) = 0.78$, $p = 0.97$, the effect of trial, the

effect of block, and trial by block interaction, respectively). In our analysis, the effects of these within-block changes were mitigated by using a control task condition with errorless periods matched to errors based on their within-block position (see above). We also made sure that the assumption of homogeneous distribution of errors within and across blocks is not violated.

**Neural substrates mediating error processing**. The main effect of errors, versus rest, is shown in Fig. 4a (see also Table 1). Significant activity increases are evident within a cluster

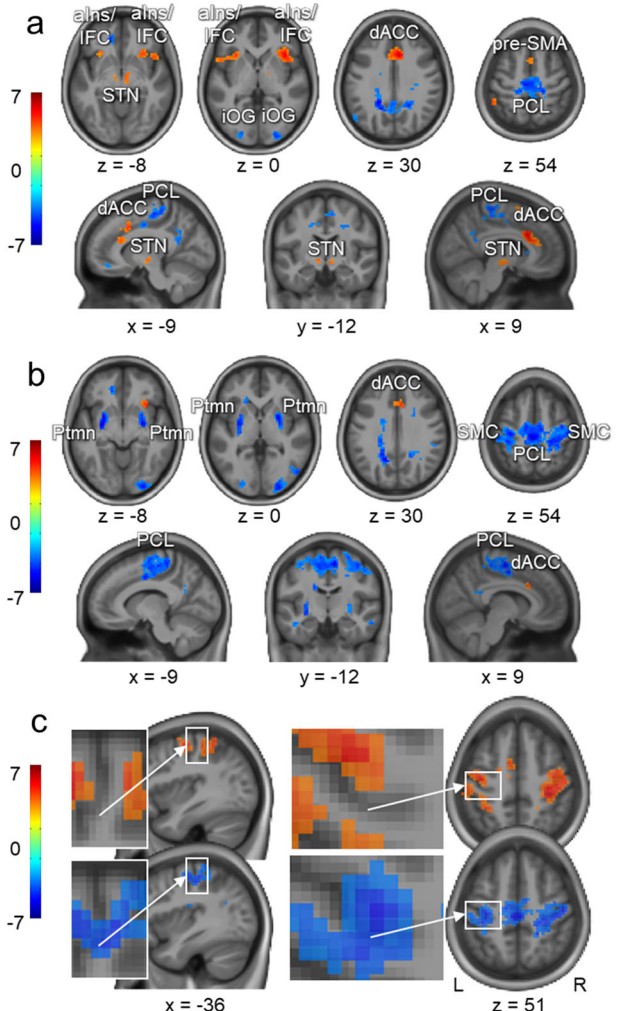

**Fig. 4 Changes in activity during errors and position-matched sequences. a** Activation maps with main effect of errors (i.e., errors versus rest). **b** Activation map with areas exhibiting significant changes in BOLD signals during errors versus their position-matched sequences (i.e., control error-free condition). **c** Regions within the sensorimotor and premotor cortices that exhibit significant activity increases during the control condition, versus rest (i.e., main effect of position-matched sequences; for additional cortical and subcortical areas see Supplementary Fig. 5), and significant activity decreases during errors versus the control condition (i.e., the same contrast as in **b**). Note that sensorimotor hand area within the left hemisphere (i.e., ipsilateral to the performing hand; the enlarged area) is not significantly engaged during the control condition, but undergoes suppression during errors. Activation maps were thresholded at $p < 0.001$. Color bars represent $t$ values. L/R - left/right hemisphere, dACC - dorsal anterior cingulate cortex, pre-SMA - pre-supplemental motor area, aIns - anterior insula, IFC - inferior frontal cortex, STN - subthalamic nucleus, PCL - paracentral lobule, SMC - sensorimotor cortex, Ptmn - putamen.

encompassing the medial prefrontal cortex, namely the dACC and pre-supplementary motor area (pre-SMA). Additional increases are observed within the aIns and inferior frontal cortex (IFC), as well as the subthalamic nucleus (STN), bilaterally. By contrast, significant activity decreases are observed within the paracentral lobule and medial parts of the pre- and postcentral gyri that correspond to the regions within the primary sensorimotor cortex (SMC) representing lower body parts. Activity decreases are also evident within the precuneus, right caudate nucleus, and posterior parts of the occipital lobe. An almost identical pattern of increased and decreased activity, versus rest, is observed when errors' onsets are shifted to the first wrong key (Supplementary Fig. 4).

The BOLD responses during errors were further estimated against those evoked during their position-matched sequences (i.e., control error-free condition; Fig. 4b and Table 1). This analysis reveals that during errors, the dACC exhibits significantly higher activity levels, indicating specific involvement of this region in error processing. Activity within the pre- and postcentral gyri, on the other hand, was significantly lower during errors. This error-specific suppression extends to the lateral parts of the SMC, including the hand knob—an anatomical landmark for the sensorimotor hand area[41]—and is bilateral (i.e., not specific to the performing hand) and widespread (Fig. 4c). Finally, at the subcortical level, error-specific decreases are observed within the putamen, bilaterally.

**Temporal characteristics of error-specific changes**. To characterize temporal dynamics in neural activity associated with error processing, we assessed changes across trials during the periods with errors relative to those without errors (Fig. 5 and Table 2); BOLD responses estimated separately for each condition, versus rest, are shown in Fig. 6. Results using statistical models with actual trial duration (Supplementary Fig. 6) - these models may be more sensitive to variable trial lengths than models with zero duration used in our main analyses - are consistent with the results reported below.

Activity increases within the dACC, pre-SMA, and aIns/IFC—the regions that are significantly activated during the error (Fig. 4a)—are already evident during the last correct trial preceding the error. This elevated activity does not change significantly when the error occurs, but drops down sharply immediately afterward, reaching near-zero values with successful performance recovery. (Figs. 5a and 6a, and Table 3(1)). We found no evidence that activity levels within these regions are significantly modulated by time spent on each error phase (Supplementary Fig. 7). This pattern of results suggests that the salience network, namely the dACC and aIns/IFC, responds instantaneously to flaws in the ongoing (or intended) action, hence, broadcasting salience PES. Similar dynamics within the pre-SMA, on the other hand, may indicate engagement of the rapid stopping mechanism that is presumably mediated via projections to the STN[32]. Indeed, slightly elevated activity at the error onset followed by a marginal decline is also observed within the STN. However, these effects fail to be significant. A similar activity pattern is exhibited by the substantia nigra (SN)/ventral tegmental area complex, with a bigger cluster located within the right hemisphere.

Activity within the NAc, bilaterally, also significantly decreases during the post-error performance recovery (Figs. 5b and 6b, and Table 3(2)). However, as opposed to the salience network, this ventral striatal region does not exhibit significant increases at the error onset so that by the time the task performance is restored, its activity levels are significantly lower than during the error-free control condition. This negative effect is rather prolonged and does not vanish by the second post-error sequence. Similar

**Table 1 Main effect of errors.**

| Area | Errors versus rest | | | | t | p Value | Errors versus matching sequences | | | | t | p Value |
|---|---|---|---|---|---|---|---|---|---|---|---|---|
| | MNI coordinates | | | Cluster size (# of voxels) | | | MNI coordinates | | | Cluster size (# of voxels) | | |
| | x | y | z | | | | x | y | z | | | |
| **(1) Error-related increases** | | | | | | | | | | | | |
| Anterior cingulate cortex[1] | 6 | 21 | 30 | 307 | 5.70 | <0.001$_{FWE}$ | 6 | 21 | 30 | 73 | 4.67 | 0.016$_{FWE}$ |
| SMA[2] | 3 | 12 | 54 | ‖ | 3.62 | ‖ | — | — | — | — | — | — |
| Right insular cortex[4] | 39 | 21 | 0 | 187 | 5.37 | <0.001$_{FWE}$ | — | — | — | — | — | — |
| Right inferior orbital frontal gyrus | 48 | 18 | −6 | ‖ | 4.39 | ‖ | — | — | — | — | — | — |
| Left insular cortex[3] | −30 | 21 | −3 | 113 | 4.53 | 0.002$_{FWE}$ | — | — | — | — | — | — |
| Left superior temporal pole | −51 | 12 | −3 | ‖ | 4.39 | ‖ | — | — | — | — | — | — |
| Left superior orbital frontal gyrus | −24 | 15 | −15 | ‖ | 4.27 | ‖ | — | — | — | — | — | — |
| Right subthalamic nucleus[6] | 6 | −12 | −9 | 18 | 3.93 | 0.003$_{FWE^*}$ | — | — | — | — | — | — |
| Left subthalamic nucleus[5] | −9 | −12 | −6 | 10 | 3.84 | 0.003$_{FWE^*}$ | — | — | — | — | — | — |
| **(2) Error-related decreases** | | | | | | | | | | | | |
| Left paracentral lobule[14] | −6 | −21 | 54 | 449 | 5.11 | <0.001$_{FWE}$ | −3 | −24 | 54 | 1959 | 5.77 | <0.001$_{FWE}$ |
| Right paracentral lobule | 9 | −27 | 63 | ‖ | 4.62 | ‖ | 6 | −30 | 57 | ‖ | 4.70 | ‖ |
| Left postcentral gyrus | −18 | −36 | 63 | ‖ | 4.51 | ‖ | −36 | −30 | 54 | ‖ | 4.87 | ‖ |
| Right postcentral gyrus | 21 | −30 | 63 | ‖ | 4.20 | ‖ | 45 | −21 | 48 | ‖ | 5.29 | ‖ |
| Left precentral gyrus[12] | — | — | — | — | — | — | −33 | −21 | 51 | ‖ | 5.28 | ‖ |
| Right precentral gyrus | — | — | — | — | — | — | 39 | −12 | 57 | ‖ | 4.69 | ‖ |
| Right SMA | — | — | — | — | — | — | 9 | −9 | 57 | ‖ | 5.22 | ‖ |
| Right dorsal putamen[19] | — | — | — | — | — | — | 24 | 3 | 0 | 139 | 5.69 | <0.001$_{FWE}$ |
| Right ventral putamen[21] | — | — | — | — | — | — | 27 | −6 | −9 | ‖ | 5.67 | ‖ |
| Left ventral putamen[20] | — | — | — | — | — | — | −27 | −6 | −9 | 162 | 5.48 | <0.001$_{FWE}$ |
| Left dorsal putamen[18] | — | — | — | — | — | — | −27 | −6 | 3 | ‖ | 4.95 | ‖ |
| Left precuneus | −6 | −57 | 21 | 574 | 4.79 | <0.001$_{FWE}$ | — | — | — | — | — | — |
| Middle cingulate cortex | 0 | −45 | 33 | ‖ | 4.26 | ‖ | — | — | — | — | — | — |
| Right caudate nucleus | 21 | 0 | 21 | 64 | 5.04 | 0.028$_{FWE}$ | 21 | 3 | 18 | 73 | 4.63 | 0.016$_{FWE}$ |
| Right inferior occipital gyrus | 27 | −93 | 0 | 77 | 4.69 | 0.012$_{FWE}$ | 24 | −93 | −3 | 151 | 5.80 | <0.001$_{FWE}$ |

Labeling clusters (the most significant local maxima for each area) obtained from activation maps thresholded at $p < 0.001$ using Automated Anatomical Labeling (AAL)[78]. The numbers in square brackets indicate areas of interest used for ROI analyses, as listed in Table 3. ‖—areas within the same cluster as area listed above, $p_{FWE}$—cluster-level FWE-corrected $p$ values over the entire brain volume, $p_{FWE^*}$—peak-level FWE-corrected $p$ values over the volume of the relevant structure using the 7T probabilistic atlas of the basal ganglia[80].

pattern of error-related changes is observed within the midline thalamic nuclei (mThlms).

Regions within the task-related network, on the other hand, exhibit changes in the opposite direction. Specifically, activity within the SMC—a region that exhibits lateralized activation during the task—reaches its minimum at the error onset and is gradually restored during the subsequent performance recovery with no evidence for lateralization (Figs. 5c and 6c, and Table 3 (3)). Similar transient error-specific decreases are observed within the dorsal premotor and parietal cortices and, at the subcortical level, within the dorsal and ventral putamen, bilaterally. Finally, changes in activity within the bilateral cerebellar lobule IV–VI are also significant. The neural dynamics within these task-activated cerebellar segments are similar to the ones observed in other motor regions. However, the magnitude of these changes is relatively small.

**Dissociation in neural dynamics between the two conditions**. To estimate to what degree the dissociation in neural activity patterns derived from the comparisons between the two conditions is, indeed, specific to errors, we conducted an additional regions of interest (ROIs) analysis using another set of position-matched sequences ("Methods" section). BOLD responses estimated for the initial and new set, versus rest, are shown in Fig. 6d–f (set 1 and set 2, respectively). None of the ROIs exhibit significant differences when the BOLD signals between the two sets are compared ($|t(48)| < 1.48$, $p > 0.15$), underpinning the specificity of the effects reported above to errors.

**Reproduction and replication study**. To estimate the reliability of our findings, we analyzed data of a subgroup of participants ($N = 28$) who were not only trained, but also retested on the same motor sequence 2 h later ("Methods" section). The analysis of the training session allowed us to determine to what degree our results are reproducible in a smaller sample, whereas data acquired separately during the retest session were used to conduct a replication study. Behavioral results derived from these analyses are shown in Supplementary Figs. 8 and 9. They are in line with the findings of our main study suggesting that error processing during continuous motor sequence production is associated with the significant drop in the performance speed. The onset of this slowing precedes the first wrong keypress and may indicate proactive error detection. We also reproduced and replicated our fMRI results (Fig. 7 and Supplementary Fig. 10) showing that during either session, error commission is associated with the instantaneous recruitment of the salience network and pre-SMA, and lingering disengagement of the NAc. A widespread suppression within the sensorimotor network specific to errors was also observed.

## Discussion

In real-life situations, errors during ongoing skilled motor performance are generally detected immediately and corrected without any externally introduced feedback. Such ability is presumably mediated via highly efficient brain mechanisms dedicated to performance monitoring and prediction error signaling. Here, we provide evidence to this theoretical assertion showing

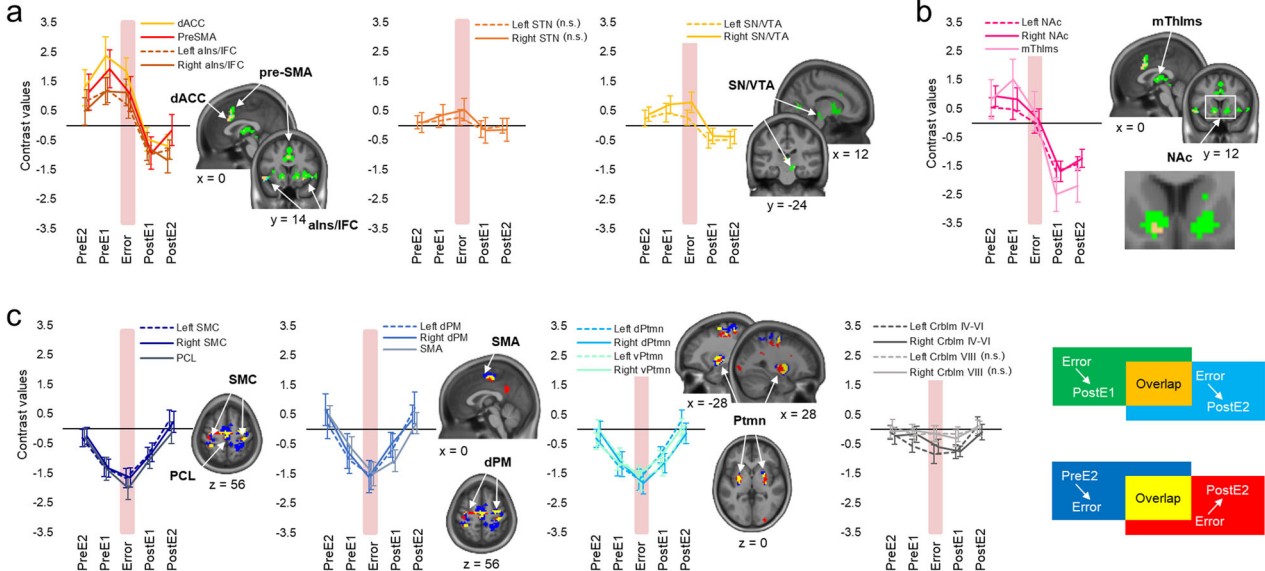

**Fig. 5 Temporal characteristics of changes in activity during error processing.** Areas with **a** increased activity during errors, **b** decreased activity during post-error trials, and **c** decreased activity during errors, respectively. Activity levels are estimated relative to the control error-free task condition. The results of the whole-brain analyses, thresholded at $p < 0.001$ (significant peaks and statistics are summarized in Table 2), are shown as colored clusters: green and cyan—areas where activity levels significantly decrease by the first and the second correct trial immediately after errors, respectively; blue—areas exhibiting decreased activity during error processing; red—areas where activity levels significantly increase with performance recovery. Data points represent mean change in activity (contrast values) for each ROI immediately before, during, and immediately after errors relative to the corresponding position-matched trials of the error-free condition (i.e., zero values along the x-axis); positive/negative values indicate activity increases/decreases during periods with errors relative to those without errors. Coordinates and statistics for each ROI are reported in Table 3. n.s. - no significant effect. dACC - dorsal anterior cingulate cortex, pre-SMA - pre-supplementary motor area, aIns/IFC - anterior insula/inferior frontal cortex, STN - subthalamic nucleus, SN/VTA - substantia nigra/ventral tegmental area complex, NAc - nucleus accumbens, mThlms - midline thalamic nuclei, SMC - sensorimotor cortex, PCL - paracentral lobule, dPM - dorsal premotor cortex, SMA - supplementary motor area, d/vPtmn - dorsal/ventral putamen, Crblm - cerebellum. PreE2 and PreE1—two consecutive sequences immediately before the error, PostE1 and PostE2—two consecutive sequences immediately after the error. Error bars represent standard error of the mean (SEM).

that movement errors during self-guided skilled motor routine are associated with significant neural changes in brain regions implicated in computations of salience and valence PES. Specifically, error commission was associated with instantaneous recruitment of the salience network and lingering disengagement of the NAc. The validity and reproducibility of these findings were confirmed using an additional set of data; we found a similar pattern of results applying the same analytical approach. Our results, thus, suggest that computations of salience and valence PES during skilled motor behavior are implemented by functionally distinct neural circuits. These circuits operate on different timescales and, therefore, may differentially contribute to immediate corrective effort and longer-term adaptive processes (i.e., learning).

Behaviorally, we observed that error commission was associated with a dramatic drop in performance speed. We replicated these results using an additional set of data (Supplementary Figs. 8 and 9), hence providing consistent support to the notion that slowing in performance constitutes a reliable behavioral signature for error detection during self-guided skilled motor routines[29]. The errors coincided with instantaneous recruitment of the salience network, including the dACC—the most studied region within the error detection system in the human brain[7,8,14].

The dACC has a diverse response profile and, in addition to its sensitivity to errors, is involved in cognitive control, conflict detection, reward processing, and somatic pain among other functions[42,43]. Seeking for a unified theoretical account, it has been proposed that the dACC plays a central role in detecting salience by estimating discrepancy between the predicted and actual outcomes[4,44]. This notion is supported by evidence that dACC in

humans and related medial frontal areas in animals consistently respond to erroneous predictions about forthcoming events[19,45,46]. These responses constitute unsigned salience PES—that is, they are sensitive to the degree of the salience/surprise estimated by the mismatch between the predicted and actual outcome, but do not indicate whether the outcome is better or worse than predicted. Such limited information conveyed via salience PES has the advantage of minimal neural resources and time required for their computations and processing. Accordingly, regions involved in processing of salience PES must exhibit positive responses not only to unexpected success or reward, but also to unpredicted outcome with negative consequences. In the current study, the instantaneous error-specific increases within the salience network are in concordance with such unsigned prediction error account, and thus appear to constitute a neural signature for error detection.

It has been proposed that the dACC may generate salience PES not only retroactively in response to the unpredicted outcome, but also proactively in situations with increased error likelihood[4,19]. This view suggests that the dACC tracks the probability of errors based on the estimation of situational factors, such as task difficulty, stimulus features, etc. When the error likelihood is increased, the proactive PES in the dACC may drive the brain toward desirable outcome by enacting inhibition of inappropriate task representations[47] and/or selective activation of more appropriate ones . It is also possible, however, that the dACC may generate proactive alerting signals in situations with fixed external conditions when the probability of errors unexpectedly increases due to internal causes. For example, during continuous motor action well-defined in advance and carried out in a stimulus-free mode, the salience

**Table 2 Changes in activity immediately before and after errors.**

| Area | MNI coordinates x | y | z | Cluster size (# of voxels) | t | p Value | MNI coordinates x | y | z | Cluster size (# of voxels) | t | p Value |
|---|---|---|---|---|---|---|---|---|---|---|---|---|
| (1) Error-specific activity increases | PreE1 < errors | | | | | | Errors > PostE1 | | | | | |
| Left anterior cingulate cortex | — | — | — | — | — | — | −6 | 18 | 30 | 174 | 4.51 | <0.001$_{FWE}$ |
| Right anterior cingulate cortex | — | — | — | — | — | — | 6 | 27 | 30 | ‖ | 4.37 | ‖ |
| SMA | — | — | — | — | — | — | 0 | 15 | 51 | ‖ | 4.34 | ‖ |
| Left anterior insular cortex | — | — | — | — | — | — | −30 | 21 | 0 | 122 | 4.66 | <0.001$_{FWE}$ |
| Left superior temporal pole | — | — | — | — | — | — | −51 | 9 | −3 | ‖ | 4.01 | ‖ |
| Right anterior insular cortex | — | — | — | — | — | — | 33 | 18 | −3 | 503 | 5.31 | <0.001$_{FWE}$ |
| Right superior temporal pole | — | — | — | — | — | — | 57 | 12 | −3 | ‖ | 3.67 | ‖ |
| Right thalamus[11] | — | — | — | — | — | — | 3 | −15 | 9 | ‖ | 4.71 | ‖ |
| Left anterior ventral putamen[9] | — | — | — | — | — | — | −12 | 12 | −9 | ‖ | 5.52 | ‖ |
| Right anterior ventral putamen[10] | — | — | — | — | — | — | 15 | 15 | −3 | ‖ | 5.26 | ‖ |
| Left substantia nigra[7] | — | — | — | — | — | — | −9 | −24 | −15 | 2 | 3.19 | 0.023$_{FWE}$* |
| Right substantia nigra[8] | — | — | — | — | — | — | 12 | −24 | −12 | 32 | 4.06 | 0.001$_{FWE}$* |
| Right calcarine cortex | — | — | — | — | — | — | 24 | −63 | 3 | 107 | 4.71 | 0.001$_{FWE}$ |
| | PreE2 < errors | | | | | | Errors > PostE2 | | | | | |
| Left anterior cingulate cortex | — | — | — | — | — | — | −3 | 18 | 33 | 52 | 3.94 | 0.042$_{FWE}$ |
| Right anterior cingulate cortex | — | — | — | — | — | — | 6 | 27 | 30 | ‖ | 3.59 | ‖ |
| SMA | — | — | — | — | — | — | 0 | 18 | 45 | ‖ | 3.37 | ‖ |
| Right insular cortex | — | — | — | — | — | — | 30 | 21 | −9 | 105 | 5.45 | 0.001$_{FWE}$ |
| Right calcarine cortex | — | — | — | — | — | — | 24 | −63 | 3 | 68 | 4.26 | 0.014$_{FWE}$ |
| (2) Error-specific activity decreases | PreE1 > errors | | | | | | Errors < PostE1 | | | | | |
| Left postcentral gyrus | −27 | −36 | 63 | 66 | 4.26 | 0.016$_{FWE}$ | — | — | — | — | — | — |
| Left precentral gyrus | −18 | −27 | 60 | ‖ | 4.10 | ‖ | — | — | — | — | — | — |
| Left caudate nucleus | −18 | −15 | 24 | 71 | 4.79 | 0.011$_{FWE}$ | — | — | — | — | — | — |
| Right supramarginal gyrus | 60 | −21 | 24 | 130 | 4.92 | <0.001$_{FWE}$ | — | — | — | — | — | — |
| Right Rolandic operculum | 45 | −21 | 21 | ‖ | 3.97 | ‖ | — | — | — | — | — | — |
| Left postcentral gyrus | −63 | −21 | 33 | 122 | 4.23 | <0.001$_{FWE}$ | — | — | — | — | — | — |
| Left supramarginal gyrus | −60 | −24 | 18 | ‖ | 3.85 | ‖ | — | — | — | — | — | — |
| Left Rolandic operculum | −45 | −30 | 18 | ‖ | 3.36 | ‖ | — | — | — | — | — | — |
| | PreE2 > errors | | | | | | Errors < PostE2 | | | | | |
| Left postcentral gyrus | −30 | −39 | 63 | 931 | 5.83 | <0.001$_{FWE}$ | −36 | −27 | 48 | 385 | 4.35 | <0.001$_{FWE}$ |
| Left precentral gyrus | — | — | — | — | — | — | −21 | −15 | 60 | ‖ | 4.16 | ‖ |
| Right SMA | 3 | −24 | 57 | ‖ | 4.77 | ‖ | 12 | −18 | 48 | ‖ | 3.89 | ‖ |
| Left SMA | −3 | −6 | 57 | ‖ | 4.18 | ‖ | −9 | −12 | 51 | ‖ | 4.38 | ‖ |
| Left middle cingulate cortex | — | — | — | — | — | — | −9 | −3 | 42 | ‖ | 5.26 | ‖ |
| Left paracentral lobule | −18 | −27 | 63 | ‖ | 3.90 | ‖ | −3 | −21 | 54 | ‖ | 3.91 | ‖ |
| Left inferior parietal gyrus | — | — | — | — | — | — | −33 | −42 | 54 | ‖ | 4.64 | ‖ |
| Left superior frontal gyrus | −24 | −9 | 63 | ‖ | 4.36 | ‖ | — | — | — | — | — | — |
| Right precentral gyrus | 33 | −21 | 63 | ‖ | 4.35 | ‖ | 30 | −12 | 51 | 155 | 4.65 | <0.001$_{FWE}$ |
| Right postcentral gyrus | 36 | −39 | 60 | ‖ | 3.89 | ‖ | 36 | −27 | 54 | ‖ | 4.58 | ‖ |
| Right superior frontal gyrus | 30 | −9 | 60 | ‖ | 4.56 | ‖ | — | — | — | — | — | — |
| Right superior parietal gyrus | 24 | −51 | 57 | ‖ | 3.80 | ‖ | — | — | — | — | — | — |
| Left dorsal putamen | −27 | −6 | 3 | 131 | 4.18 | <0.001$_{FWE}$ | −30 | −6 | 0 | 89 | 5.54 | 0.004$_{FWE}$ |
| Left ventral putamen | — | — | — | — | — | — | −30 | −9 | −9 | ‖ | 4.32 | ‖ |
| Left amygdala | −24 | −3 | −12 | ‖ | 5.47 | ‖ | — | — | — | — | — | — |
| Left Rolandic operculum | −48 | −3 | 6 | ‖ | 3.60 | ‖ | — | — | — | — | — | — |
| Right dorsal putamen | 24 | 3 | 0 | 71 | 4.85 | 0.011$_{FWE}$ | 30 | −9 | 3 | 86 | 5.20 | 0.004$_{FWE}$ |
| Right amygdala | 27 | 0 | −12 | ‖ | 4.74 | ‖ | — | — | — | — | — | — |
| Left caudate nucleus | −18 | −18 | 24 | 70 | 4.84 | 0.012$_{FWE}$ | — | — | — | — | — | — |
| Right supramarginal gyrus | 54 | −36 | 30 | 62 | 5.04 | 0.020$_{FWE}$ | — | — | — | — | — | — |
| Right Rolandic operculum | 45 | −21 | 21 | ‖ | 3.75 | ‖ | — | — | — | — | — | — |
| Right posterior insular cortex | 39 | −15 | 15 | ‖ | 3.55 | ‖ | — | — | — | — | — | — |
| Left Rolandic operculum | −48 | −21 | 18 | 79 | 4.25 | 0.007$_{FWE}$ | — | — | — | — | — | — |
| Left superior temporal gyrus | −60 | −27 | 15 | ‖ | 4.05 | ‖ | — | — | — | — | — | — |

Labeling clusters (the most significant local maxima for each area) obtained from activation maps thresholded at $p < 0.001$ using Automated Anatomical Labeling (AAL)[78]. Activity levels during continuous periods of errorless performance juxtaposed to the matching sequences were used as a baseline. The numbers in square brackets indicate areas of interest used for ROI analyses, as listed in Table 3. ‖—areas within the same cluster as area listed above. $p_{FWE}$—cluster-level FWE-corrected $p$ values over the entire brain volume; $p_{FWE}$*—peak-level FWE-corrected $p$ values over the volume of the relevant structure using the 7T probabilistic atlas of the basal ganglia[80].

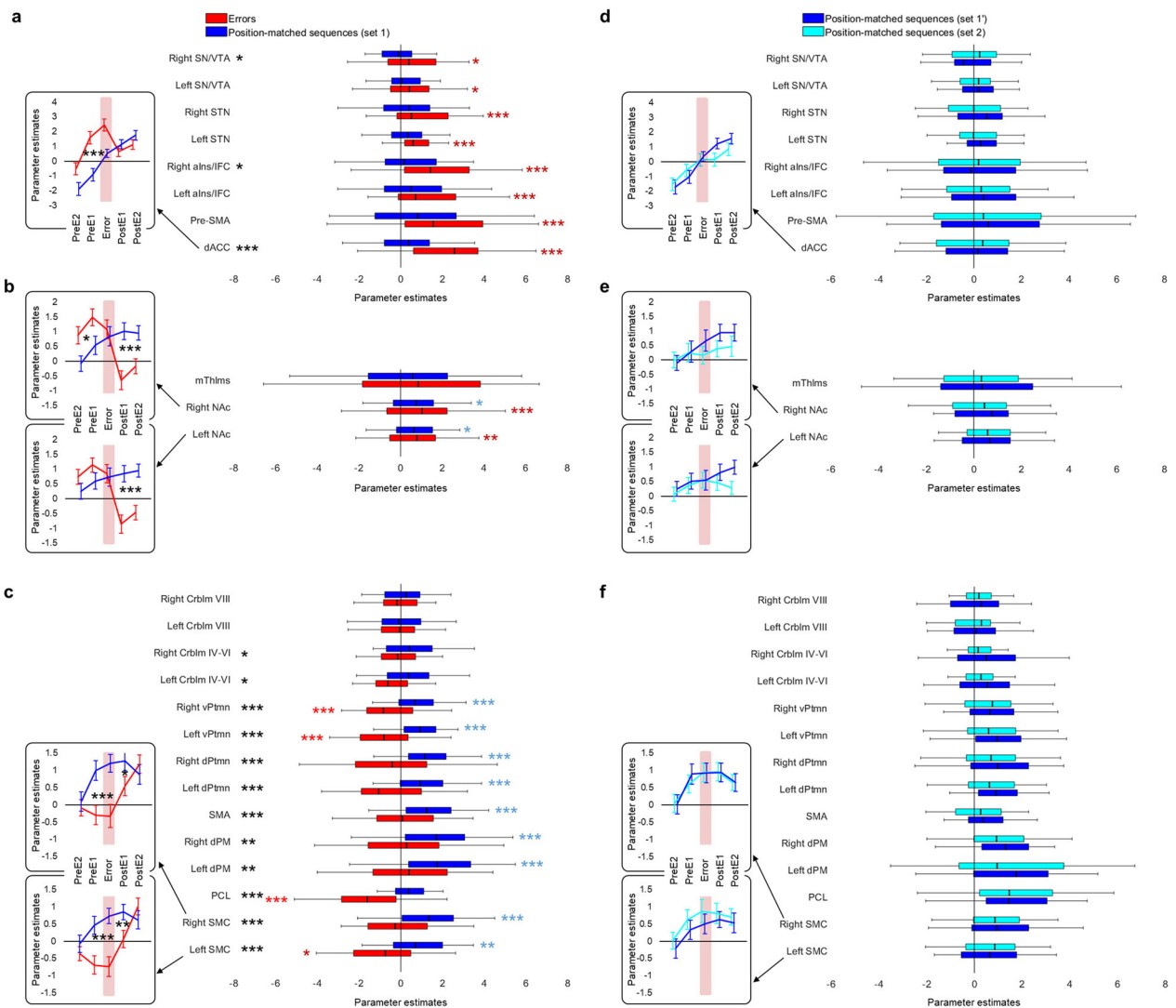

**Fig. 6 Activity levels during errors and their position-matched sequences versus rest.** Parameter estimates at the onset of the errors (red) and position-matched sequences (blue), versus rest (zero values along the *x*-axis), extracted for regions of interest (ROIs; Table 3) with **a**, **d** error-specific activity increases, **b**, **e** post-error activity decreases, and **c**, **f** error-specific activity suppression. Black asterisks indicate significant differences between errors and position-matched sequences; red and blue asterisks indicate significant activity increases/decreases during errors and position-matched sequences, versus rest, respectively. *, ** and *** - significant results at 0.05, 0.01 and 0.001 level respectively; in the later case, the results are significant after Bonferroni correction for 25 ROIs included in the analysis (*p* < 0.002). The abbreviations for brain regions are the same as in Fig. 5. PreE2 and PreE1—two consecutive sequences before the error, PostE1 and PostE2—two consecutive sequences after the error. Error bars represent standard error of the mean (SEM).

network may respond to the unexpected flaws in the intended action even before the negative consequences of these flaws become evident behaviorally and the actual error occurs. Although with caution because of the limitations imposed by the low temporal resolution of the fMRI signal, we speculate that in the current study, the activity peak within the salience network shifted toward the pre-error phase (Figs. 5a and 7a) could potentially indicate such proactive salience PES. These alerting signals may forecast error commission during skilled motor performance enacting inhibitory control even before the erroneous movement is fully executed[20]. Indeed, our results indicate that slowing in performance occurred as early as the error onset, even before the first wrong key was pressed (Fig. 3b, c), and was also associated with rapid recruitment of the pre-SMA and widespread suppression of the sensorimotor circuits, including bilateral primary sensorimotor cortices and putamen. Previous human and animal studies suggest that the pre-SMA is crucially involved in outright suppression of the initiated

action[11,48]. This medial prefrontal region is a node within the fast STN-mediated hyperdirect stopping mechanism that has global (i.e., not specific to the currently engaged task representations) suppressive effects on the motor system[32]. Therefore, it is plausible that slowing in performance at the error onset observed in the current study reflects an unsuccessful attempt of the brain to prevent the forthcoming error by imposing global motor suppression. The efficiency of this effort may depend on the capacity of the salience network to anticipate forthcoming error and generate alerting PES in advance, before the erroneous movement is actually carried out.

In addition to being salient events, movement errors during skilled motor routine also constitute a sudden violation of predicted success. They indicate a failure to achieve the desired outcome and, therefore, may trigger negative valence PES within the reward circuits[49]. These negative signals can then be used to assign blame to actions that led to this failure[1]. Our results indicate that the NAc—a structure within the ventral

**Table 3 Parameter estimates and statistics derived from the analyses of the regions of interest (ROIs).**

| Area | | Contrast used to define ROI | MNI coordinates | | | PreE2 | | | | PreE1 | | | | Error | | | | PostE1 | | | | PostE2 | | |
|---|---|---|---|---|---|---|---|---|---|---|---|---|---|---|---|---|---|---|---|---|---|---|---|---|
| | | | x | y | z | mean | SEM | p Value | | mean | SEM | p Value | | mean | SEM | p Value | | mean | SEM | p | | mean | SEM | p |
| **(1) Error-specific increases** | | | | | | | | | | | | | | | | | | | | | | | | |
| (1) | dACC | Errors > rest | 6 | 21 | 30 | 1.292* | 0.616 | 0.041 | ↗*** | 2.369*** | 0.643 | 0.001 | ↗*** | 1.850*** | 0.447 | 0.000 | ↗*** | −0.462 | 0.400 | 0.253 | n.s. | −0.695 | 0.400 | 0.089 |
| (2) | Pre-SMA | Errors > rest | 3 | 12 | 54 | 1.146 | 0.614 | 0.068 | n.s. | 1.929** | 0.650 | 0.005 | n.s. | 1.077 | 0.585 | 0.071 | ↗* | −0.931 | 0.556 | 0.101 | n.s. | −0.152 | 0.526 | 0.773 |
| (3) | Left IFC/aIns | Errors > rest | −30 | 21 | −3 | 0.859* | 0.355 | 0.019 | n.s. | 1.207** | 0.436 | 0.008 | n.s. | 0.680 | 0.433 | 0.123 | ↗*** | −0.993** | 0.321 | 0.003 | n.s. | −0.812* | 0.331 | 0.018 |
| (4) | Right IFC/aIns | Errors > rest | 39 | 21 | 0 | 0.485 | 0.463 | 0.301 | n.s. | 1.169* | 0.453 | 0.013 | n.s. | 1.124* | 0.458 | 0.018 | ↗*** | −0.702 | 0.476 | 0.147 | n.s. | −1.175** | 0.421 | 0.008 |
| (5) | Left STN | Errors >rest | −9 | −24 | −15 | 0.113 | 0.226 | 0.618 | n.s. | 0.157 | 0.220 | 0.479 | n.s. | 0.279 | 0.264 | 0.297 | n.s. | −0.123 | 0.280 | 0.663 | n.s. | −0.006 | 0.262 | 0.983 |
| (6) | Right STN | Errors > rest | 12 | −24 | −12 | 0.098 | 0.335 | 0.772 | n.s. | 0.370 | 0.339 | 0.281 | n.s. | 0.541 | 0.374 | 0.154 | ↗* | −0.163 | 0.434 | 0.709 | n.s. | −0.135 | 0.392 | 0.732 |
| (7) | Left SN/VTA | Errors > PostEI | −9 | −24 | −15 | 0.253 | 0.279 | 0.370 | n.s. | 0.439 | 0.320 | 0.177 | n.s. | 0.276 | 0.253 | 0.281 | n.s. | −0.492 | 0.264 | 0.069 | n.s. | −0.477 | 0.278 | 0.093 |
| (8) | Right SN/VTA | Errors > PostEI | 12 | −24 | −12 | 0.392 | 0.245 | 0.116 | n.s. | 0.721* | 0.297 | 0.019 | n.s. | 0.794* | 0.352 | 0.029 | ↗*** | −0.348 | 0.272 | 0.207 | n.s. | −0.365 | 0.237 | 0.130 |
| **(2) Post-error decreases** | | | | | | | | | | | | | | | | | | | | | | | | |
| (9) | Left NAc | Errors > PostEI | −12 | 12 | −9 | 0.552 | 0.322 | 0.093 | n.s. | 0.481 | 0.363 | 0.191 | n.s. | 0.020 | 0.378 | 0.959 | ↗*** | −1.676*** | 0.358 | 0.000 | n.s. | −1.416*** | 0.247 | 0.000 |
| (10) | Right NAc | Errors > PostEI | 15 | 15 | −3 | 0.928** | 0.360 | 0.013 | n.s. | 0.828* | 0.400 | 0.044 | ↘** | 0.115 | 0.383 | 0.765 | ↗*** | −1.686*** | 0.360 | 0.000 | n.s. | −1.230*** | 0.315 | 0.000 |
| (11) | mThlms | Errors > PostEI | 3 | −15 | 9 | 0.828 | 0.684 | 0.232 | n.s. | 1.527* | 0.691 | 0.032 | ↘** | 0.423 | 0.682 | 0.538 | ↗*** | −2.497*** | 0.591 | 0.000 | n.s. | −2.174*** | 0.587 | 0.001 |
| **(3) Error-specific decreases** | | | | | | | | | | | | | | | | | | | | | | | | |
| (12) | Left SMC | Errors < matching seq | −33 | −21 | 51 | −0.305 | 0.287 | 0.293 | ↗* | −1.286*** | 0.317 | 0.000 | n.s. | −1.668*** | 0.325 | 0.000 | ↗** | −0.904** | 0.279 | 0.002 | ↗*** | 0.296 | 0.347 | 0.398 |
| (13) | Right SMC | Errorless blocks > rest | 36 | −21 | 51 | −0.234 | 0.287 | 0.418 | ↗*** | −1.333*** | 0.307 | 0.000 | n.s. | −1.641*** | 0.334 | 0.000 | ↗* | −0.799* | 0.318 | 0.015 | ↗** | 0.239 | 0.351 | 0.500 |
| (14) | PCL | Errors < matching seq | −3 | −24 | 54 | −0.272 | 0.304 | 0.375 | ↗** | −1.357*** | 0.387 | 0.001 | ↘* | −2.004*** | 0.375 | 0.000 | ↗** | −1.018** | 0.318 | 0.002 | ↗** | −0.126 | 0.367 | 0.734 |
| (15) | Left dPM | Errorless blocks > rest | −27 | −9 | 57 | 0.179 | 0.491 | 0.717 | ↗** | −1.028* | 0.457 | 0.029 | n.s. | −1.549*** | 0.473 | 0.002 | n.s. | −0.634 | 0.496 | 0.207 | ↗** | 0.732 | 0.540 | 0.182 |
| (16) | Right dPM | Errorless blocks > rest | 27 | −6 | 60 | 0.667 | 0.540 | 0.223 | ↗** | −0.721 | 0.590 | 0.228 | ↘* | −1.590*** | 0.538 | 0.005 | ↗** | −0.519 | 0.420 | 0.223 | n.s. | 0.340 | 0.480 | 0.482 |
| (17) | SMA | Errorless blocks > rest | −6 | −3 | 57 | 0.450 | 0.353 | 0.208 | ↗*** | −0.632 | 0.377 | 0.100 | ↘** | −1.503*** | 0.392 | 0.000 | n.s. | −1.057** | 0.363 | 0.005 | ↗** | 0.220 | 0.354 | 0.536 |
| (18) | Left dPtmn | Errors < matching seq | −27 | −6 | 3 | −0.300 | 0.298 | 0.319 | ↗*** | −1.218*** | 0.360 | 0.001 | ↘* | −1.701*** | 0.360 | 0.000 | ↗** | −0.861** | 0.273 | 0.003 | ↗*** | 0.323 | 0.327 | 0.328 |
| (19) | Right dPtmn | Errors < matching seq | 24 | 3 | 0 | −0.135 | 0.406 | 0.740 | ↗*** | −1.177* | 0.419 | 0.007 | ↘** | −1.825*** | 0.363 | 0.000 | ↗*** | −1.145** | 0.320 | 0.001 | ↗*** | −0.132 | 0.347 | 0.705 |
| (20) | Left vPtmn | Errors < matching seq | −27 | −6 | −9 | 0.007 | 0.322 | 0.984 | ↗*** | −0.956** | 0.339 | 0.004 | ↘* | −1.722*** | 0.339 | 0.000 | ↗*** | −1.276*** | 0.301 | 0.000 | ↗*** | −0.138 | 0.277 | 0.621 |
| (21) | Right vPtmn | Errors < matching seq | 27 | −6 | −9 | −0.397 | 0.275 | 0.155 | ↗** | −1.107*** | 0.279 | 0.000 | ↘* | −1.515*** | 0.280 | 0.000 | ↗*** | −0.957** | 0.252 | 0.000 | ↗*** | 0.085 | 0.268 | 0.752 |
| (22) | Left Crblm IV–VI | Errorless blocks > rest | −18 | −51 | −24 | −0.094 | 0.233 | 0.688 | ↗* | −0.551* | 0.225 | 0.018 | n.s. | −0.839* | 0.317 | 0.011 | ↗* | −0.758* | 0.239 | 0.003 | ↗** | 0.117 | 0.315 | 0.711 |
| (23) | Right Crblm IV–VI | Errorless blocks > rest | 24 | −57 | −27 | −0.221 | 0.290 | 0.449 | n.s. | −0.163 | 0.268 | 0.546 | ↘* | −0.592* | 0.257 | 0.026 | ↗*** | −0.748** | 0.187 | 0.000 | ↗*** | −0.095 | 0.265 | 0.721 |
| (24) | Left Crblm VIII | Errorless blocks > rest | −18 | −60 | −51 | 0.086 | 0.256 | 0.738 | n.s. | −0.171 | 0.234 | 0.469 | n.s. | −0.140 | 0.279 | 0.617 | n.s. | −0.160 | 0.255 | 0.534 | n.s. | 0.026 | 0.256 | 0.919 |
| (25) | Right Crblm VIII | Errorless blocks > rest | 24 | −57 | −54 | −0.063 | 0.220 | 0.778 | n.s. | −0.020 | 0.200 | 0.920 | n.s. | −0.136 | 0.266 | 0.611 | n.s. | −0.298 | 0.279 | 0.291 | n.s. | 0.093 | 0.237 | 0.698 |

Regions of interest (ROIs) were defined as spheres with the radius of 6 mm; smaller radius of 4 mm was used to define ROIs within the basal ganglia and the thalamus. The mean contrast values of parameter estimates and corresponding statistics are provided for each ROI at the onset of two adjacent sequences immediately before the error (PreE2, PreE1), at the onset of the error and at the onset of two adjacent sequences immediately after the error (PostE1, PostE2). The contrast values were extracted using activity levels during continuous correct performance at the corresponding time (i.e., sequence position) within the block as a baseline. ↗—significant increases between consecutive sequences. ↘—significant decreases between consecutive trials.
n.s. - non-significant results, dACC - dorsal anterior cingulate cortex, pre-SMA - pre-supplementary motor area, aIns/IFC - anterior insula/inferior frontal cortex, SN/VTA - substantia nigra/ventral tegmental area complex, NAc - nucleus accumbens, mThlms - midline thalamic nuclei, SMC - sensorimotor cortex, PCL - paracentral lobule, dPM - dorsal premotor cortex, SMA - supplementary motor area, d/vPtmn - dorsal/ventral putamen, Crblm - cerebellum, PreE2 and PreE1 - two consecutive sequences before the error, PostE1 and PostE2 - two consecutive sequences after the error.
*, **and ***, significant results at 0.05, 0.01 and 0.001 level, respectively; p < 0.002 indicates significant result after Bonferroni correction for 25 ROIs included in the analysis

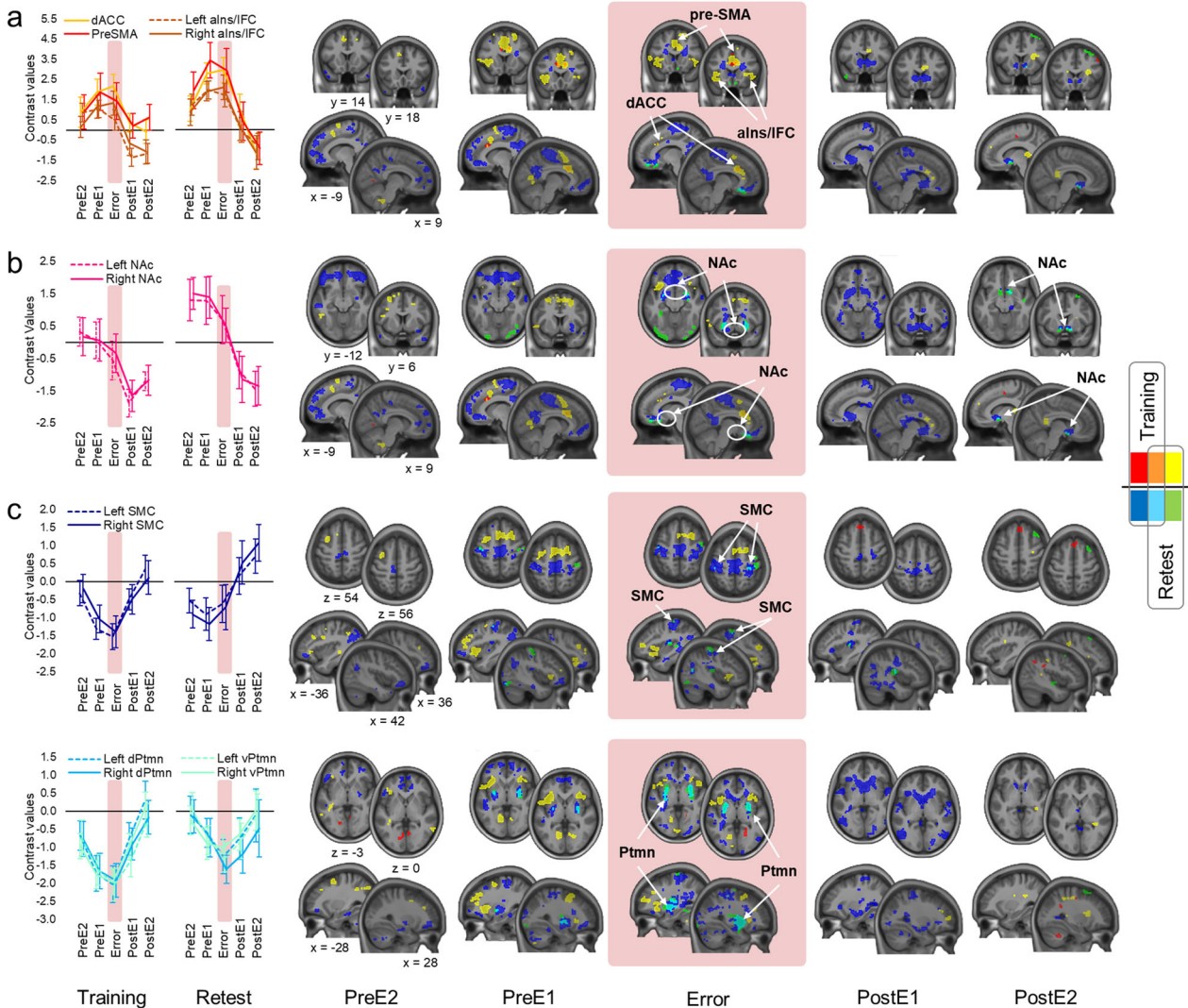

**Fig. 7 Reproduction and replication study.** Areas with a increased activity during errors, **b** decreased activity during post-error performance recovery, and **c** decreased activity during errors. We used the same regions of interest (ROIs) as in our main study (Table 3). Activity levels in both ROI-based and whole-brain analyses are estimated relative to the control error-free task condition (zero values along the *x*-axis). Data points represent mean contrast values. The results of the whole-brain analyses were thresholded at $p < 0.005$. Red/blue and yellow/green clusters—activity increases/decreases associated with error processing during the training and retest, respectively; orange/cyan clusters—overlap. The abbreviations for brain regions are the same as in Fig. 5. PreE2 and PreE1—two consecutive sequences before the error, PostE1 and PostE2—two consecutive sequences after the error. Error bars represent standard error of the mean (SEM).

striatum and a valence center in the mammalian brain[21,22,50] (for the recent review, see ref. [51])—may be involved in such retroactive negative signaling. This dopamine-innervated subcortical region did not exhibit significant changes in activity at the error onset. However, following error commission, its activity levels significantly decreased and, by the time the task performance was restored, were significantly lower than during errorless performance (Figs. 5b and 6b; see also Fig. 7 and Supplementary Fig. 10). Such disengagement of the NAc is incompatible with the salience or action initiation account[24,25] (Supplementary Note 1). Instead, it may indicate a pause in dopamine transients in response to errors[50,52,53] and, to the best of our knowledge, constitutes the first fMRI evidence of negative valence PES elicited by movement errors during self-guided skilled motor routine.

The disengagement of the NAc was rather prolonged and did not vanish by the second post-error sequence. This observation is consistent with lingering reduction in NAc dopamine

concentration in response to aversive stimuli in animals[54]. Dips in dopamine levels are thought to affect synaptic plasticity in the indirect striatal pathway[55]. Traditionally, this pathway has been considered as a NoGo route that suppresses actions[56,57] in a behaviorally specific, context-dependent manner[58]. It is presumably implicated in aversive memory in mice[59,60], disfavor of unrewarded targets in monkeys[61,62], and learning from negative outcomes in humans[63,64]. Accordingly, the lingering disengagement of the NAc observed in the current study may indicate enduring learning-from-error signals that impose suppression of neural populations representing the erroneous action. The selective punishment of undesirable representations during correct performance immediately after the error may refine selective activation of neural populations that can reliably produce the target sequence of movements, thereby promoting formation of a dedicated pathway for the effortless and stereotyped execution of the practiced motor skill[65,66]. Thus, negative PES elicited retroactively in response to errors have the

potential to facilitate learning. This idea is supported by previous studies showing that, in humans, punishment-based monetary feedback leads to significantly faster online motor responses[67] and accelerates error-based motor learning[68].

Previous evidence also indicates that motor learning can benefit from negative feedback even without subsequent monetary loss, but only if it is directly linked to actual errors in performance[68]. This finding resonates with the conjecture that negative valence PES may facilitate learning via selective suppression targeting neural populations that represent the erroneous action. If error-related disengagement of the NAc reported here mediates such selective suppression, it should not have negative effect on performance once it has been recovered. Indeed, lingering disengagement of the NAc following errors did not prevent participants from rapidly accelerating their tapping speed to the levels similar to errorless performance (Fig. 3d and see also Supplementary Fig. 9). Thus, rather than imposing a negative impact, neural processes underlying such disengagement may, instead, facilitate post-error performance recovery, and possibly learning, by enacting prolonged selective suppression of neural populations that provoked the error.

In the current study, additional activity decreases within the striatum associated with errors were observed in its dorsolateral (sensorimotor) portions, namely the putamen. These decreases, however, occurred during early phases of error processing and rapidly disappeared once the task performance was recovered. Such temporal dynamics, which are equivalent to those exhibited by other regions engaged in the motor task (Fig. 5c), emphasize the role of the sensorimotor striatum in habitual behavior and skilled motor routines[69,70] over its putative involvement in processing valence[71] (see also Supplementary Note 1).

We conclude that during skilled motor performance the key brain regions within the salience- and reward-responsive circuits play distinct, yet complementary roles. Our results indicate that the salience network responds instantaneously to movement errors, hence, broadcasting salience PES. The reward-responsive circuit, on the other hand, may signal negative valence. Here, these negative PES were presumably expressed as lingering disengagement of the NAc, which persisted over several post-error trials, but did not interfere with the task performance. These findings suggest that the salience- and reward-responsive circuits play distinct roles in immediate interruption of the erroneous action and subsequent performance recovery by generating fast salience and delayed valence PES, respectively. Salience PES may be also delivered proactively in response to flaws in the intended action even before the actual error occurs, whereas valence PES may contribute to longer-term adaptive processes, hence facilitating learning.

## Methods

**Ethics statement**. All participants gave their written informed consent to take part in the study, which was approved by the Research ethics board of the RNQ (Regroupement Neuroimagerie Québec). All procedures were in accordance with the approved guidelines and regulations. Participants were compensated for their participation.

**Participants**. The current report is based on the analyses of data collected during the training session from an fMRI experiment published elsewhere[27]. The sample included 55 healthy young (mean age: 24.1 ± 3.5 years, 34 females) right-handed[72] volunteers recruited by local advertisements to participate in the study. Participants reported no history of medical, neurological, or psychiatric disease. None of the participants were taking medications at the time of testing. Also, none received formal training on a musical instrument or as a typist. All participants had normal quality of sleep, as assessed by the Pittsburgh Sleep Quality Index questionnaire[73] and the St. Mary Hospital questionnaire[74]. Data from one participant was excluded due to excessive head movement. From the 54 participants whose data were analyzed in the current study, 52 of them completed at least one performance block (out of 14) without any error (mean = 7.02 errorless

blocks, SEM = 0.40). Errors that met the criteria for analyses (see below) were detected within the data of 49 participants (mean = 4.14 errors, SEM = 0.30).

**Motor sequence task**. The experiment was carried out using a finger motor sequence learning task. The task was programmed in Matlab R2014a (The Mathworks, Inc., Natick, MA), using Cogent 2000 developed by the Cogent 2000 team at the FIL, and the ICN and Cogent Graphics developed by John Romaya at the LON at the Wellcome Department of Imaging Neuroscience (http://www.vislab.ucl.ac.uk/cogent.php). Participants were scanned using fMRI, while practicing the sequential finger tapping task[26]. Lying supine in the scanner, participants were instructed to tap a five-element sequence of finger movements on a four-key response pad using their left (nondominant) hand (Fig. 1). The sequence (4–1–3–2–4) was introduced to participants using numbers from 1 to 4 that corresponded to the four fingers of their left hand (excluding the thumb) from the index to the little finger, respectively. Full explicit introduction of the sequence was given before the scanning session began. Participants were asked to memorize the sequence and to accurately reproduce it. The session was initiated only after the sequence was successfully performed three times in a row, without any error. Participants performed the task in a stimulus-free condition (no feedback) being asked to look at the fixation cross, and to tap the sequence repeatedly as fast and accurately as possible. In case of occasional errors, they were instructed to continue with the task as smoothly as possible from the beginning of the sequence. The training session consisted of 14 successive blocks of practice comprising 60 keypresses, i.e., equivalent to 12 repetitions of the correctly performed and competed sequence, separated by 15-s periods of rest. During rest, participants were asked to remain still, without moving their fingers, and look at a fixation cross. The change in color of the fixation cross, from red to green and from green to red, indicated the beginning (GO cue) and the end (STOP cue) of each performance block, respectively. Participants' performance was registered saving the code number (i.e., 1, 2, 3, or 4) and time of each keypress.

**fMRI data acquisition**. Functional MRI series were acquired using a 3.0 T TIM-TRIO scanner system (Siemens, Erlangen, Germany), equipped with a 32-channel head coil. T2*-weighted axial fMRI images were obtained with a gradient echo-planar sequence using interleaved acquisition mode in ascending direction (TR = 2650 ms, TE = 30 ms, FA = 90°, FoV = 220 × 220 mm$^2$, matrix size = 64 × 64 × 43, voxel size = 3.4 × 3.4 × 3 mm$^3$, 10% inter-slice gap). T1-weighted sagittal 3D MP-RAGE structural images were also obtained (TR = 2300 ms, TE = 2.98 ms, TI = 900 ms, FA = 9°, FoV = 256 × 256 mm$^2$, matrix size = 256 × 256 × 176, voxel size = 1 × 1 × 1 mm$^3$).

**Behavioral data analyses**. An error comprised all wrong keypresses that violated the predetermined order (i.e., sequence) or incomplete sequences with missing keys, between two correct trials (i.e., instances of correctly performed and completed sequences). To be included in the analysis, the error should have met several criteria. It should have been preceded and followed by at least three correct trials in a row; if the error was followed by two correct trials at the very end of the block, it was also considered (Fig. 1a). In that way, errors of interest were isolated from each other and, together with their adjacent trials, formed separate error periods. Such separation allowed us to track the evolution of error-related changes throughout all phases of error processing, including the error commission and the subsequent performance recovery. It also ruled out the possibility that errors included in the analysis would coincide with block transients—activity bursts when transitioning between cognitive states at the onset and offset of the task blocks[38]. To avoid analyzing extremely short or recurring errors, the error also should have lasted longer than 0.5 s and comprised less than 20 keys (i.e., the number of keys in four correctly pefomed and completed sequences).

To account for possible performance deterioration and changes in activity within the block ("Results" section), we detected continuous periods with at least seven (or six if the period was at the very end of the block) correct trials, matched them to error periods by their within-block position, and used them as a control task condition. Potential differences in the statistical power between the two conditions (i.e., periods with and without errors), were minimized by including the same number of periods in each condition. To do so, we applied a pseudo-random position-matching procedure. For each period with errors, we first detected all periods without errors at the same within-block position and then randomly chose only one. Errors, for which position-matched periods without errors were missing, were excluded. Since group-level analyses were performed on individual outcomes derived from comparisons between these equally scaled task conditions, such an approach also allowed us to mitigate between-individual differences in the number of errors.

The within-block position of each period was determined by its fourth trial, which during periods with and without errors constituted an error and so-called position-matched sequence, respectively. As we mentioned before, each block could comprise a maximum of 12 correct trials. (i.e., sequences). Since each error of interest should have been preceded by at least three and followed by at least three (or two if it was the last error in the block) correct trials in a row, it could be located at any position between 4 and 10 within the block. This position was

determined using the following formula:

$$\text{ROUND}\left(\frac{i-1}{k}\right) + 1. \tag{1}$$

ROUND—rounding to the nearest integer, $i$—the within-block index of the first trial key, and $k$—the sequence length (i.e., the number of keys within the sequence; in the current study, $k = 5$).

We used transition duration, i.e., a time interval between two consecutive keypresses, as the main behavioral measure that reflected speed. To account for changes in performance across blocks, transition durations were converted into percentage relative to the mean transition duration within all correct trials on a block-by-block basis.

The data were analyzed using Statistical Package for the Social Sciences (SPSS Statistics for Windows, Version 24.0; IBM Corp., Armonk, NY). All analyses were designed as within-subject comparisons using either paired $t$ test (two-tailed) or repeated-measures analyses of variance (ANOVA) with trial type/period, phase (trial onset, within-trial, trial offset), within-block position and/or block as within-subject factors. Corrections for violations of sphericity were made using the Greenhouse–Geisser adjustment.

**fMRI data preprocessing.** The structural and functional images were converted to Neuroimaging Informatics Technology Initiative format using MRIcron (University of South Carolina). Preprocessing and statistical analysis of the data were carried out with SPM12 (http://www.fil.ion.ucl.ac.uk/spm/software/spm12/; Wellcome Trust Centre for Neuroimaging, London, UK) operating under Matlab R2014a (The Mathworks, Inc., Natick, MA).

During initial preprocessing, functional volumes were realigned using a least squares approach and a six-parameter (rigid body) spatial transformation to correct for movement-related variance. Following segmentation and skull-stripping of the structural data, functional images were coregistered to the individual skull-stripped 3D anatomical image and normalized to the Montreal Neurological Institute (MNI) space, using parameters obtained from the segmentation procedure. The normalized functional images were resampled to voxel dimensions of 3 mm³ and spatially smoothed with an isotropic Gaussian kernel with a full-width at half-maximum of 6 mm to improve the signal-to-noise ratio.

**fMRI data analyses.** Statistical analyses of fMRI time series consisted in a two-stage summary statistics model[75]. In the first stage, BOLD signal changes were estimated for each subject independently using a fixed-effect general linear modeling (GLM). We used a mixed block/event-related design[40] to separate transient activity related to trials of interest (errors/sequences) from sustained task-related activity during continuous motor sequence practice (peaks and statistics for mean task-related activity are summarized in Supplementary Table 1). Each model comprised covariate for performance periods represented as a boxcar function, time-locked to the onset and duration of each block, and covariates for trials represented as a stick function (i.e., zero duration) time-locked to the trial onset. The boxcar function effectively separates epochs of interest (i.e., blocks) from the periods of rest allowing to estimate mean brain activity that is sustained throughout the task. The stick function, on the other hand, is well suited to capture neural changes time-locked to the event onset, and is less sensitive to differences in activity due to variations in the duration and/or processing time of events[76]. Thus, the mixed design approach allowed us to assess changes in neural activity closer associated with trial onsets, while minimizing biases from varying trial durations inherently present in our design and accounting for the mean task-related activity.

To minimize the effect of block onset transients[38], an additional covariate represented as a stick function time-locked to the task onset (GO cue) was included in the models. All covariates were convolved with a hemodynamic impulse response function. Movement parameters derived from realignment of the fMRI time series were also included as covariates of no interest. A high-pass filter of 128 s was used to remove low-frequency noise. Serial correlations in fMRI signal were estimated through a restricted maximum likelihood algorithm, using a first-order autoregressive plus white noise model. High collinearity between covariates for closely spaced (in time) trials may result in highly variable parameter estimates due to the procedure of GLM parameter estimation that automatically removes the effects of the shared variability[77]. To preserve variance explained by each covariate, BOLD signal changes associated with adjacent trials in a block were estimated using separate models. As a result, 12 models, for each trial within errorless blocks, were specified to estimate within-block changes as captured by the stick function (Supplementary Fig. 3). Additional five models, for errors as well as two adjacent trials before and after errors, were determined to estimate neural changes immediately before, during, and immediately after errors. In these models, covariates for corresponding position-matched trials composing periods without errors were also included and used as a control condition. Details about selection criteria for periods with and without errors can be found above in the behavioral data analyses section. Following parameter estimation, the $t$-contrasts (i.e., univariate linear combinations of parameter estimates) were defined to test the main effect of task blocks (i.e., sustained task-related activity) and of each trial type (i.e., errors and sequences), versus rest. To assess changes in activity specific to errors, BOLD signals evoked during periods with errors were contrasted with

BOLD signals evoked during periods without errors. These contrasts were calculated on a trial-by-trial basis comparing parameter estimates between each pair of position-matched trials.

In the second stage, the resulting contrast images ($t$-maps) were carried forward to the random effects analyses to assess the consistency of effects between subjects (group-level analyses). The inferences about sustained task-related activity and changes time-locked to trials (i.e., errors and sequences) were done using a one-sample $t$ test. To get insight into temporal characteristics of error-related network, comparisons between error-specific activity changes immediately before, during, and after errors were performed using a one-way within-subject ANOVA. Unless otherwise stated, activation maps were thresholded at $p \leq 0.001$ and overlaid on the mean structural image of all participants using Functional Imaging Visualization Environment toolbox for SPM (FIVE, http://mrtools.mgh.harvard.edu). All images are oriented according to the neurological convention (i.e., the left side of the image corresponds to the left side of the brain).

Statistical inferences were performed at the cluster level using $p$ values family-wise error rate (FWE)-corrected for multiple comparisons over the entire brain. Clusters of brain activation were labeled according to anatomical automatic labeling[78]. The boundaries of the motor and premotor cortices were defined based on the human motor area template[79]. Localization of subcortical activation within basal ganglia and the brain stem was performed using the 7T probabilistic atlas of the basal ganglia[80].

ROIs were defined within the brain areas involved in motor sequence production[34], error processing[81,82], and inhibitory control[83] based on activation maps of task- and error-related activity (task blocks versus rest, errors versus rest, errors versus control error-free task condition). In that way, the ROIs within these networks were localized independently of the whole-brain analyses of changes over the pre-, during, and post-error trials. Additional areas were identified based on activation maps of significant changes during the post-error performance recovery (errors versus first post-error correct trials). ROIs were defined as spheres with the radius of 6 mm; smaller radius of 4 mm was used to define ROIs within the basal ganglia and the thalamus. Parameter estimates and contrast values of parameter estimates were extracted from sphered ROIs centered at the local maxima of thresholded maps resulted from group-level analyses using MarsBar toolbox for SPM[84]. The extracted values were introduced to SPSS (Version 24.0; IBM Corp., Armonk, NY) for further analyses.

Due to summation effects across consecutive closely spaced (in time) trials[85], the temporal accuracy of neural changes associated with error processing is biased by the adjacent trials. Therefore, we expected that onset of error-specific changes in BOLD signal would be shifted in time. In addition, the sampling time of 2.65 s used to acquire fMRI data, which is longer than an average duration of a single sequence, imposed additional limitations on timing precision so that changes associated with error commission could be actually captured in the BOLD signal of the preceding trial, as well as endure over a few consecutive trials. Given this limitation, the inferences about differential contribution of particular networks to error processing were made based on shape and peak/dip latency of temporal activity patterns determined by contrast values of relative changes immediately before, during, and immediately after errors.

**Statistics and reproducibility.** We estimated the specificity and reliability of our results in several ways. First, we generated another set of position-matched sequences (set 2). To do so, we applied the pseudo-random matching procedure to the initial set of such sequences (set 1) to ensure that the two sets are different. Twenty sequences from set 1 without position-matched sequences in set 2 were excluded, thereby leaving for the analysis 183 sequences within each set with 3.389 ± 0.29 (mean ± SEM) position-matched sequences on average for each individual. We used this new set to estimate to what degree the dissociation in neural activity patterns derived from the comparisons between the two conditions, i.e., periods with and without errors is specific to errors.

Next, we reproduced our findings on a sub-sample of participants ($N = 28$). Two hours after the initial training (the data analyzed in our main study), these participants also took part in another separate experimental session. The design of this retest session was identical to the design of the training session (Fig. 1a), with the only exception that participants performed the sequence with the keyboard turned upside down—the EGO condition as described by Albouy and colleagues[27]. We used the data acquired during this additional session to conduct a replication study.

**Reporting summary.** Further information on research design is available in the Nature Research Reporting Summary linked to this article.

## Data availability

All data are available upon reasonable request from the corresponding authors. All source data underlying the graphs and charts presented in the manuscript are available via Figshare: https://doi.org/10.6084/m9.figshare.13114202.v1.

## Code availability

The computer code is available upon reasonable request from the corresponding authors.

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

## Acknowledgements

This study was supported by grants from the Canadian Institutes of Health and Research (CIHR; MOP 97830) as well as the Ministry of Economic Development, Innovation and Exportation of Quebec (MDEIE; PSR-SIIRI-704) to J.D. E.G. was supported in part by postdoctoral fellowship provided by the Fonds de recherche du Québec – Santé (FRQ-S).

## Author contributions

G.A. and J.D. designed the experiment. G.A. performed the experiment. E.G. analyzed the data. E.G. and J.D. wrote the original draft. G.A. and O.L. reviewed and edited the manuscript. O.L. provided expertise and feedback.

## Competing interests

The authors declare no competing interests.
