## [Peer Review File · Communications Biology]

Reviewers' comments:

Reviewer #1 (Remarks to the Author):

This paper reports detailed analysis of behavioural and fMRI data in a sequence production task, gathered as a preliminary part of another previously published report.

The aim was to measure neural responses to rare errors in the key-stroke sequences, and to determine whether there are network differences in the error-free vs erroneous motor performance, in error detection and in salience and valence processing (the former driven by errors; the latter considering the sign of the error). Because the number of spontaneous errors was low, data from a sub-group of the experiment were also analysed, as a "replication".

It is an interesting approach, and in principle a good design, and I think the questions are of wide interest. But I am not especially enthusiastic for several reasons.

First the task requires sequences to be as fast as possible, and so individual keystrokes are well beyond the temporal resolution of the fMRI analysis. To some extent this has been mitigated by considering whole sequences of 5 keystrokes at a time, but this greatly reduces the interest of the results, as it seems critical to assess error onset at the level of single keystrokes, not sequences.

Second, the instructions to participants were that when they made errors they should effectively restart from the beginning of the sequence. The long increase in duration may well reflect this process of resetting and reinitiating the sequence. Hence it seems appropriate to compare post-error performance with initiation of the sequences at the start of each block, and not with performance mid-block.

Third, errors were rare – only 203 across nearly 50 participants, and so there must be considerable variability – some participants making only one, some possibly making many more. This may lead to bias in the results, towards both behavioral and fMRI effects driven by only a few participants. It may well affect the statistics, and the t-tests used to compare behavioral data may be compromised by uneven variance. At a minimum, so analysis of the distribution of errors across the population should be presented. The attempt to mitigate against these low numbers with a replication study is good – but a sub-sample of the original pool of participants is not a valid replication. An independent sample of new participants is needed.

Fourth I am puzzled but not sure I fully understand how the fMRI data is modeled. The authors state the errors are modeled a zero duration. I guess this will bias the fitting towards very short error sequences, and away from the longer multi-sequence events? Could this effect lead to the apparent reduction in fMRI signal in error sequences?

I also am not sure I fully understand the behavioral measures. Error duration is from end of a correct sequence to start of a correct sequence, if I understand it. Hence one would expect a clear order effect with longer duration for later elements in the sequence, if participants successful stop and reset the sequence as instructed. But this was not seen. In addition, when comparing % change in transitions to mean % change in duration of pre and post sequences, it seems one is comparing a single keystroke change to mean change over 5 keystrokes, and so it is unsurprising the latter are smaller effects.

Finally, the design rests on spontaneous errors, all of which are bad. So, I cannot see how the data can support an argument on valence, which would seem to need separation of positive and negative prediction outcomes. There are no unexpected positive outcomes.

Reviewer #2 (Remarks to the Author):

The authors set out to identify the neural correlates of 'salience' (unsigned) and 'valence' (signed) Prediction Errors related to reward. The contributions of this manuscript are twofold: first, although signed and unsigned PEs have been observed throughout the brain, there is ongoing debate as to how/where these signals may overlap/interact. In particular, the authors focus on dACC is apt; dACC is the subject of much debate along these lines, with various authors suggesting that the error signals in dACC are signed, unsigned, or a mixture of the two. Given the array of findings that can support any of these positions, it is a valuable to directly investigate the nature of error signals in dACC, especially in contrast to other areas that are more reliably implicated in signed PEs (nucleus accumbens).

Second, the authors attempt to disentangle these PEs in the area of motor control. dACC is generally acknowledged to be critical in cognitive control, to which motor control is related. By relating their motor control paradigm to processes and functions frequently studied in the context of cognitive control, the manuscript aims to generalize findings from one line of research (cognitive control) to a separate (although closely related) line.

To do this the authors adopt a finger-tapping task in which subjects are asked to memorize a five-digit sequence in which they are to tap their fingers. During the task proper, subjects tap their finger in a continuous, self-paced fashion, and error sequences are compared to non-error sequences in both behavioral and fMRI analyses.

On the whole this study addresses an important and relevant issue in the field of cognitive/motor control, i.e., distinguishing the nature of different error signals in the brain, especially w/r/t salience and value. The approach taken to answering this question seems sensible, and the major results appear to be of sufficient interest for the readership. However, there are a few major concerns I would like to see addressed before this manuscript can be accepted.

MAJOR

1) Some aspects of the experiment design are unclear or ambiguous. For example, I guess the term 'matching sequence' means instances when the subject correctly reproduced the learned sequence of key presses (?), but it is not clearly defined anywhere in the manuscript that I could find, and it seems from text on lines 298-299 errorless performance is not the same thing as the matching sequence... this is one example of the design needs to be made more clear.

2) The analyses that contribute to fig. 5 (A,B,C), pg. 15, seem to run afoul of one of the classical neuroimaging fallacies. Specifically, the authors claim that, using errorless performance as a baseline, 'salience' network regions are specifically engaged on error and NOT matching sequences. It appears they base this on a significant difference from the BOLD response in those regions to the errorless baseline. However, just because the BOLD signal is different from baseline in one condition, and not different from baseline in another condition, this does not indicate that the BOLD signal in condition 1 is different from the BOLD signal in condition 2. I note that they do not do this in the analyses supporting figure 3, so perhaps they could make it clear that they are not basing all their claims on the fig. 5. analyses.

3) The authors note increased activity in the salience network prior to the occurrence of an error; this does not seem to fit well with their story about the 'instantaneous' recruitment of the salience network in response to errors; if the salience network responds instantly to error, why should it be active during correct performance preceding an error. I think there are ways one could support an unsigned PE story for these periods, but it doesn't appear that the authors have attempted to do so. I would like a more thorough treatment of how increased pre-error activity relates to the underlying theory supporting the authors predictions.

4) Given the authors' explicit aim to dissociate value PEs from salience PEs, especially in the realm of control, it seems appropriate to acknowledge recent work that likewise attempts to dissociate value and surprise signals in control, specifically Vassena et al., 2020, Nature Human Behavior. Full disclosure: I am an author on that paper, but it legitimately seems relevant to this study.

5) From your behavioral analyses, it appears that there is some amount of response slowing in the PreE1 periods; dACC activity is frequently correlated with time on task (e.g., grinband et al., 2010). It appears that the fMRI analyses do not incorporate parametric modulators that might be used to model response time or sequence time. It could be possible that the increased salience network activity observed in preE periods might be related to increases in RT. They authors should probably re-rerun the analyses with RT included as a modulator to ensure that their data could not be explained as deriving from longer sequence durations.

6) This task doesn't appear to carry an explicit reward manipulation, i.e., subjects are not given any incentive (monetary or otherwise) to perform the sequences correctly. I don't necessarily think that such an incentive is absolutely required, but given the long history of studying reward/valence PEs with explicit rewards, the authors should spend some time justifying why the signals they observe in, e.g., NAcc are in fact reflecting reward PEs and not some other signal.

MINOR

the word 'acclaim' does not mean what you think it means... I think you want the word 'claim' in those cases.

Reviewer #3 (Remarks to the Author):

Communications Biology Review

The authors had subjects repeated a simple motor task involving repeating a sequence of key presses many times over, while being scanned in an fMRI. The subjects occasionally made errors in the sequence, but were not given related feedback. Thus, any knowledge of the error came from endogenous, not exogenous sources. During sequences in which the subject made an error, the speed of performance was slower. In addition, the dorsal anterior cingulate, pre-supplementary motor area, inferior frontal cortex, subthalamic nucleus, substantia nigra and subthalamic nucleus all showed activity increases, whereas paracentral lobule and sensorimotor cortex showed activity decreases. In addition, performance slowed even more following the error, and activity dropped in the medial thalamus and nucleus accumbens. The authors list many, many effects, but the take-home message is that regions associated with negative-valence prediction errors, such as dACC increase in activity during errors, whereas regions associated with task performance, such as sensorimotor cortex showed reduced activity.

The task is interesting, the N is impressive, and the data is fairly compelling. I have only a few points.

1. The paper is difficult to understand. I have to take responsibility for some of that – it is likely that part of my difficulty is merely due to me not being smart enough, or well-versed enough in some of the subject matter. It is also partly due to the fact that English may not be the first language of the authors. Here and there are some tell-tale mistakes in word choice. For example, page 4, line 67, "upon such conditions"; page 17, line 336, "theoretical acclaim showing that movement..."; and page 18 line 363, "caution is imposed because of the low temporal resolution..." These are small language errors that I will not dwell on. I do suggest the authors have a native English speaker proofread the work.

One identifiable, general problem in the writing is that the authors often use overly long sentences, trying to cram too many pieces of information into one point. For example, the first two sentences on page 12 (lines 235-243):

"Next, we assessed temporal characteristics of error-specific changes in activity by estimating changes in BOLD responses immediately before, during and immediately after errors relative to those evoked during continuous periods of errorless performance immediately before, during and after matching sequences, respectively."

And then

"Thus, we conducted both region of interest (ROI) and whole-brain analysis applying repeated measures ANOVA approach on individual contrast values and activation maps, respectively, obtained from pairwise comparisons between estimated BOLD responses during (1) penultimate sequences before errors and before matching sequences, (2) last sequences immediately before errors and before matching sequences, (3) errors and matching sequences, (4) first sequences immediately after errors and after matching sequences, and (5) second sequences after errors and after matching sequences."

I understand that in the later sentence, lists tend to make long sentences, but I still argue that both sentences are made harder to follow by being too long. I would advise/request that the authors go through the paper carefully with the goal of breaking the longest sentences into two or more shorter, more succinct sentences. I realize that some of the points the authors are trying to make are complex and nuanced, but still, breaking such points into smaller sections with a clear progression of what needs to be understood in what order could greatly improve the readability of this work.

I also have to note that most of my points have to do with not understanding the description, so in each case, re-writing portions of the text may be the solution.

2. Exactly what point in an error was the fMRI time-locked to? It seems that when the participant got a sequence wrong at any point, that the entire sequence as labelled an error. Does that mean that the participant could get item 1 through 4 right, press the wrong key for item 5, and then the onset of the error is still locked to the onset of the first (correct) key press?

3. Related to determining errors, what happened if the participant made a single keypress error on the first item, and then instead of going to what would normally be the second item, repeated the first item correctly and then continued on. Would that make all following key presses an error? That is, how did the authors handle errors that were merely cases where the sequence was offset by one?

4. Page 8 line 170: "we further reaffirmed that slowing associated with error commission occurred as early as at the error onset, persisted during the error and reached its peak at the error offset" Is "at error onset" here the transition from the first erroneous press to the next press, or the transition from the last correct press to the first incorrect press?

5. The "matching sequences" were matched according to their within-block position, meaning an error on block 1 could be matched to an errorless sequence on block 14, is that right? There appears to be an effect across blocks on transition duration. Why would the authors match to across blocks instead of just matching to periods of errorless performance within the same block?

REVIEWER 1

REVIEWER 1

We are pleased to know that the reviewer has noticed that we used an “...interesting approach, and in principle a good design...” and that s/he thinks that “...the questions are of wide interest”. We appreciate the reviewer’s time spent on reviewing our manuscript.

We made substantial efforts to address each of the Reviewer’s concerns. We believe that this effort helped us to clarify the advantages of our approach, to improve the readability of the manuscript, and to support better our conclusions.

Reviewer

First the task requires sequences to be as fast as possible, and so individual keystrokes are well beyond the temporal resolution of the fMRI analysis. To some extent this has been mitigated by considering whole sequences of 5 keystrokes at a time, but this greatly reduces the interest of the results, as it seems critical to assess error onset at the level of single keystrokes, not sequences.

Response

The reviewer is correct that the main units of interest in our analysis were trials rather than single keypresses. If the sequence was correctly performed and completed it was considered as one correct trial. Any other combination of consecutive keypresses between two correct trials were considered as one erroneous trial, or simply an error. Such an approach suits better the low temporal resolution of the fMRI signal, which does not capture changes across single keypresses that are only a few hundreds of milliseconds apart. Thus, we agree that this approach has its limitations and does not allow assessing the precise timing of “switching” and interactions between neural mechanisms underlying different phases of error processing on a scale of milliseconds. Yet, the acquisition of the whole-brain fMRI data and the subsequent set of analyses allowed us to capture error-related changes that occur on a slower timescale at the network level in a space-resolved manner. In addition to the trial-level analysis, behavioral data were also analyzed at the level of single keypresses. In the revised version of the manuscript, an explicit explanation to our approach and its rationale is now provided in the first paragraph of the Results section as follows:

“The main units of interest in our analysis are trials, i.e., sequences and errors, comprising several keypresses (Fig. 1a). Such an approach suits better the low temporal resolution of the fMRI signal allowing us to assess neural dynamics across trials in a space-resolved manner. To get insights into the types of errors and changes in performance during errors, behavioral data were also analyzed at the level of single keypresses.

Five consecutive keypresses that followed the predetermined order (i.e., sequence) are considered as one correct trial. Any other combination of consecutive keys (or a single key), including instances of incomplete sequence, between two correct trials constitutes an error (Fig. 1b). Thus, the first wrong keypress that violated the predetermined order of the sequence could occur after up to four keys pressed correctly” (page 7, lines 105-112).

Reviewer

Second, the instructions to participants were that when they made errors they should effectively restart from the beginning of the sequence. The long increase in duration may well reflect this process of resetting and reinitiating the sequence. Hence it seems appropriate to compare post-error performance with initiation of the sequences at the start of each block, and not with performance mid-block.

Response

Indeed, we instructed participants to restart the sequence from the first key and continue with the task as smoothly as possible if they noticed that they made an error, rather than trying to correct it midstream. Once the predetermined order of keypresses is violated, trying to figure out the last correct keypress and the next key that should have been pressed may not be a trivial task. Our guidelines, however, did not require such an effort from participants; instead they were provided with a clear and easy approach to deal with errors. This explanation is now included in Supplementary Materials as a Supplementary Note.

We agree with the reviewer that there are some similarities between restarting the task after the error and initiating it after the rest (i.e., the start of the block). Both require a similar set of processes, such as (re)planning and executing the sequence. However, unlike restarting the task after an error in a self-guided manner (i.e., in the absence of any external cue) during the performance block, restarting the task after the rest at the beginning of the block was guided by the external “GO” cue. Therefore, to be able to initiate the task after the rest, participants also needed to attend

REVIEWER 1

to the “GO” cue and, once it was detected, to make a transition from “doing nothing” to “doing something”. Such exogenous attention and switching between the two cognitive states are presumably mediated via generic brain mechanisms that are not specific to the task at hand. This notion is supported by previous fMRI studies that used various tasks, designed as blocks, and showed transient non-task-specific activity bursts at block transition points, i.e., when participants switched between the rest- and task-states (Dosenbach et al., 2006; Fox et al., 2005; Konishi et al., 2001; Mechelli et al., 2003).

Nevertheless, following the reviewer’s comment, we conducted additional analysis of behavioral data comparing post-error performance with the successful task initiation at the start of the blocks. For each individual, we matched the number of the first-block sequences considered in the analysis with the number of error periods by applying a random selection procedure from all correctly initiated blocks. In this way, the statistical power to detect changes was comparable across different sequence/trial types. Performance levels for the mid-block (position-matched) sequence, which was used as a control condition in the manuscript, were also included in the analysis.

First, we compared the distribution of trials of each type (i.e., first, mid-block and post-error sequences) across blocks and found no significant results (effect of block: $F(13, 624) = 1.532, p = 0.101$; block by trial type interaction: $F(13.55, 650.47) = 1.294, p = 0.208$), which suggest non-heterogeneous and equivalent nature of trial distribution. Next, we assessed differences in performance levels between the trial types treating different trial phases (i.e., trial onset and within trial performance), separately. Both, the onset and execution of the post-error sequence differ significantly from those of the first- and mid-block sequences ($|t(48)| > 2.595, p \leq 0.013$). Thus, the post-error slowing was of a greater magnitude than slowing after rest, even though in the letter case there was a need for exogenous attention. These results are now included in Supplementary Materials as Supplementary Fig. 2 and are mentioned in the Results section as follows:

“The degree of this post-error slowing surpasses the degree of the reduced speed at the block initiation phase (Supplementary Notes, Supplementary Fig. 2) and, therefore, cannot be fully explained by the need to reinitiate the task per se” (page 10, lines 177-179).

In response to the same comment, we also reiterated our previous analysis of changes in activity at the beginning of the performance blocks (please, see the figure below); this analysis was part of the pipeline to extract within-block changes during errorless blocks shown in Supplementary Fig. 3. Specifically, we generated a group activation map with main effect of the first sequence within the blocks; for comparison, a group activation map with main effect of the first post-error sequence is also shown. For each subject, the number of events in the regressor used to assess the effect of the first-block sequence was equal to the number of errors analyzed for that subject so that the statistical power to detect changes was comparable to the statistical power to detect changes at the post-error phase. Nevertheless, the effect of the first-sequence within the block was incomparably stronger and much wider spread than the effect of the post-error sequence as indicated by the fact that we had to use higher initial threshold levels and a more restrictive correction ($p < 0.05$, FWE-corrected at the peak level) to obtain activation maps with distinct clusters for each condition. In addition, we also extracted parameter estimates from representative regions of each circuit of interest, namely the dACC, NAc and sensorimotor cortices. Except for the dACC, which exhibited comparable activity levels during any of the three sequences, activity within other regions was much stronger and robust during the first-block sequence than during the mid-block or post-error sequence. Thus, activity increases at the beginning of performance blocks was not only wide-spread, extending beyond task-related circuits, but also disproportionately high. Such transient overshoots at the task onset can be spurious, emerging by a non-physiological mechanism of slice misplacement (Renvall and Hari, 2009); this slice misplacement may explain movement-like artifacts near tissue boundaries on the activation map with the main effect of the first-block sequence shown in the figure below. Therefore, first-block sequences may not be a suitable control condition to estimate error-specific changes during the post-error sequence.

In fact, in the current study, we made a particular effort to minimize these effects of transient activity bursts at block initiation points by including “...an additional covariate represented as a stick function time-locked to the task onset (“GO” cue)...” in statistical models of fMRI signals (Methods; page 26, lines 527-528). Moreover, only isolated errors preceded and followed by at least three correct trials in a row were analyzed. **“Such an approach allowed us to minimize the overlap between the pre- and post-error intervals and to reduce the effects of transient non-specific activity bursts at task initiation and termination (Fox et al., 2005)” (Results; page 7, lines 116-118).**

(a) Activation map with main effect of the first-block sequence ($p < 0.05$, FWE-corrected). Only blocks with successful task initiation starting with correctly performed and completed sequence (S1) were included in the analysis. For each subject, the number of events in the regressor used to assess the effect of the first-block sequence was equal to the number of errors analyzed for that subject, so that the statistical power to detect changes was comparable to the statistical power to detect changes during the post-error sequence (PostE1). (b) Activation map with main effect of the first post-error sequence (PostE1) ($p < 0.001$, unc). (c-e) Parameter estimates extracted from the dACC, NAc and SMC. Green, gray and yellow columns represent group means. Error bars represent s.e.m.

REVIEWER 1

We would also like to emphasize that the post-error slowing was neither a focus of our study nor our key finding. With great recognition of the previous work that gave rise to several influential theories aiming to explain this ubiquitous phenomenon, including cognitive control theories (Botvinick et al., 2001; Ridderinkhof et al., 2004), orienting account (Notebaert et al., 2009) and activation-suppression hypothesis (Ridderinkhof, 2002), we did not intend to test and/or reconcile between these accounts in the context of the continuous motor task used in the current study. Instead, focusing on the error phase itself, “we aimed to elucidate (1) whether error processing during skilled motor performance is associated with changes in brain regions implemented in computations of salience and valence PES, and if so, (2) how these changes evolve throughout different phases of error processing” (page 5, lines 77-81). The continuous nature of the task employed in the current study allowed us to track these changes over continuous performance periods with and without errors. Direct comparisons between these two periods also allowed us to capture dynamics specifically linked to errors.

We recognize that in the previous version of the manuscript, we did not provide sufficient explanation about the rationale of using the mid-block trials composing continuous periods of errorless performance as a control condition in the main text. In the current version, such an explanation is included in the Results section as follows:

“To account for changes in performance and in neural activity associated with fatigue (Rickard et al., 2008), time-on-task (Grinband et al., 2011), and the position of the trial within blocks (see the fMRI results below), periods with and without errors were pseudo-randomly matched by their within-block position (Methods)” (page 7, lines 126-128).

Reviewer

Third, errors were rare – only 203 across nearly 50 participants, and so there must be considerable variability – some participants making only one, some possibly making many more. This may lead to bias in the results, towards both behavioral and fMRI effects driven by only a few participants. It may well affect the statistics, and the t-tests used to compare behavioral data may be compromised by uneven variance. At a minimum, so analysis of the distribution of errors across the population should be presented.

Response

We agree with the reviewer that it is essential to show the distribution of errors across participants. Initially, this information was included in Supplementary Materials. Now, it is provided in Fig. 2. Out of 49 participants, 11, 26 and 12 participants committed between 1 to 2, 3 to 5, and 6 to 7 errors, respectively, all of which met the inclusion criteria. We took several measures to account for these differences in the number of errors across participants. First, the position-matched periods with and without errors were equalized in number for each individual. Since group-level analyses were performed using within-subject comparisons between these equally scaled periods/conditions, it also allowed us to mitigate between-individual differences in the number of errors. Second, before proceeding with the behavioral analysis, we first computed a single value (i.e., mean transition duration in secs or magnitude of slowing in %) for each participant and each trial/phase of interest (i.e., errors, sequences/correct trials, pre- and post-error trials, trial onsets and offsets, etc.). These values were then used in group-level statistical tests to assess changes in performance associated with error processing. Third, during the fMRI data analysis, a single regressor was used to model all trials of each type/condition at the individual level. In that way, parameter estimates and t-maps were calculated based on values that were automatically scaled by the number of events included in each regressor. We revised our manuscript and added the following explanations to make sure that these points are clear:

“Potential differences in the statistical power between the two conditions (i.e., periods with and 130 without errors), were minimized by including the same number of periods in each condition” (page 8, lines 130-131).

“All measures were first calculated at the individual level and only then introduced to the group-level analysis” (page 10, lines 158-159).

Reviewer

The attempt to mitigate against these low numbers with a replication study is good – but a sub-sample of the original pool of participants is not a valid replication. An independent sample of new participants is needed.

Response

We share this reviewer’s view and agree that the analysis of data acquired for the sub-group of participants during the same experimental session is not a valid replication. Therefore, we conducted a replication study using another

REVIEWER 1

data set acquired for these participants during a separate session. We revised our statements in this regard making sure that it is clear. For example:

“We not only reproduced our results for a sub-group of participants from the same experimental session but also conducted a replication study using data acquired during another separate session” (page 5, lines 89-91).

The pattern of changes observed in this within-subject replication study was consistent with the findings of our main study. In the previous version of the manuscript, the results of the replication analysis were reported only in Supplementary Materials. After the revision, these results are now summarized in the main text (page 16, lines 289-299). The information about error distribution and behavioral results are shown in Supplementary Fig. 6 & 7. The fMRI results are visualized in Fig. 7 & Supplementary Fig. 8.

Reviewer

Fourth I am puzzled but not sure I fully understand how the fMRI data is modeled. The authors state the errors are modeled a zero duration. I guess this will bias the fitting towards very short error sequences, and away from the longer multi-sequence events? Could this effect lead to the apparent reduction in fMRI signal in error sequences?

Response

The reviewer is rising an important point. Indeed, various trial lengths could constitute a confounding factor in the analysis of the fMRI data. Exactly for this reason, all trials of interest were modeled as events using a stick function with zero duration. This decision was made based on previous studies that suggest that the stick function, which is sometimes also referred to as the impulse function, is particularly suited to capture neural changes time-locked to the event onset and is less sensitive to differences in activity due to variations in the duration and/or processing time of events (Grinband et al., 2008; Henson, 2006; Mechelli et al., 2003). However, given a relatively low time-resolution of the fMRI signal, it is still possible that the stick function would also capture changes associated with the subsequent trial. Such a scenario would decrease the sensitivity of our design to capture error-specific changes, especially during short errors, rather than increase the probability of false positives, as suggested by the reviewer.

Following the reviewer’s comment, in addition to details provided in the Methods section (page 26, lines 514-526), we now also include a short explanation about the fMRI signal modeling in the Results section as follows:

“To determine neural substrates mediating error processing, functional data were analyzed using a mixed block/event-related design (Visscher et al., 2003). Performance periods were modeled as blocks/epochs, whereas trials of interest (i.e., errors and sequences) were modeled as events. To minimize the effects of various trial durations, trial-related changes in the blood-oxygen-level-dependent (BOLD) signal were estimated using a stick function with zero duration (Methods)” (page 12, lines 200-203).

To address the reviewer’s concern regarding so called “multi-sequence” events – if we understand correctly the reviewer is referring to the possibility that errors may contain several attempts to restart the sequence – we performed a detailed analysis of the error keys to determine how many attempts were made to reinitiate the correct sequence within the error. Such an attempt was determined by the first sequence transition, i.e., the key combination of 4-1 (Fig. 1b; this newly-added panel also shows the distribution of errors by the number of within-error keys before and after the first wrong key was actually pressed). As it is shown in Fig. 2c, in the vast majority of cases (191 out of 203 errors), correct task performance was successfully reinitiated from the first attempt, which minimizes the possibility that slowing in performance and changes in brain activity during errors can be biased by the number of such attempts.

Finally, we also performed an additional whole-brain analysis shifting error onsets to the first wrong keypress – the difference between these two events is visualized in Fig. 1b. This analysis showed that the activation patterns almost perfectly overlap with those derived from our original analysis of error-related effects (Supplementary Fig. 4). These new results are now also mentioned in the Results section of the manuscript.

“An almost identical pattern of increased and decreased activity, versus rest, is observed when errors’ onsets are shifted to the first wrong key” (page 12, line 218-219).

Reviewer

I also am not sure I fully understand the behavioral measures. Error duration is from end of a correct sequence to start of a correct sequence, if I understand it. Hence one would expect a clear order effect with longer duration for later elements in the sequence, if participants successful stop and reset the sequence as instructed. But this was not seen.

REVIEWER 1

Response

The reviewer is correct about the way the error duration was estimated. Yet, we are not certain if we understand why “one would expect a clear order effect with longer duration for later elements in the sequence”. Previous studies on the motor sequence task, including our own (Gabitov et al., 2017, 2019a, 2019b), have shown that transitions between each two consecutive keys of the correctly executed sequence are not of the same duration. Moreover, these durations do not follow a particular order and do not necessarily increase by the later keys of the sequence. Also, in the current study, participants were instructed to perform the sequence repeatedly and continuously until the end of the performance block and were not asked to stop every time when they completed the sequence.

“The sequence was reproduced in a continuous and self-paced manner without relying on any external cue or input” (page 4, lines 63-64).

Only in the case of errors, participants were asked to “reset” their performance and to restart the sequence from its first element. As it is shown in Fig. 3b & 3c, successful performance “reset” after errors, to which we also refer as the error offset, was associated with the greatest slowing. The magnitude of such immediate post-error slowing was several times greater than during other error phases and could be a product of either or both successful interruption of erroneous action and performance reset. As we mentioned before, despite being an important, intensively studied, yet, still debatable topic in the field, a detailed investigation of the post-error slowing and its underlying processes is beyond the scoop of our study.

Reviewer

In addition, when comparing % change in transitions to mean % change in duration of pre and post sequences, it seems one is comparing a single keystroke change to mean change over 5 keystrokes, and so it is unsurprising the latter are smaller effects.

Response

Before addressing the reviewer’s comment, we would like to clarify the term “sequence” and its usage in the context of the current work. Specifically, when we use this term, we refer to a set of five consecutive keys pressed in the predetermined (sequential) order, i.e., the proper sequence. In the current study, the sequence that participants were instructed to perform was 4-1-3-2-4. Any other combination of the keypresses between two correctly performed and completed sequences was considered as n error. To make this point explicit and clear, we added the following explanation in the first paragraph of the Results section:

“Five consecutive keypresses that followed the predetermined order (i.e., sequence) were considered as one correct trial. Any other combination of consecutive keys (or a single key), including instances of incomplete sequence, between two correct trials constitutes an error” (page 7, lines 109-111).

Our understanding is that the reviewer’s comment refers to the behavioral analysis in which each error phase – error onset, within-error transitions and error offset – was treated as a separate within-subject measure; in the revised manuscript within-error transitions are further divided into transitions before and after the first wrong keypress. Indeed, the magnitude of slowing at the error onset/offset was estimated based on the mean of between trial transitions immediately before/after error, respectively. The number of such transitions across errors is lower than the number of the within-error transitions. Therefore, to get comparable estimates, in terms of the statistical power, and to account for possible variations in speed of different transition types, i.e., transitions between and within trials, we not only compared between the error phases themselves (Fig. 3b) but also assessed them relative to the corresponding phases of position-matched sequences (Fig. 3c). As we mentioned before, the number of errors and position-matched sequences considered in the analysis was exactly the same. Both analyses showed a very similar pattern of results. In the main text, results derived from the phase-by-phase comparisons between the two conditions (i.e., errors and position-matched sequences) are summarized as follows:

“Finally, we also compared the magnitude of slowing during distinct error phases with the corresponding phases of position-matched sequences (i.e., control task condition) and reaffirmed that slowing associated with error commission is, indeed, specific to errors and evident throughout all error phases (Fig. 3c)” (page 11, lines 180-182).

Reviewer

Finally, the design rests on spontaneous errors, all of which are bad.

Response

Studying spontaneous behavior is a challenging endeavor, but it is also important for the advancement of our understanding, since this is ubiquitous to real-life situations. Recognizing these challenges, we implemented the following steps. First, we performed in-depth analysis of the behavioral data providing detailed information about types of errors and their distribution (Fig. 1b & Fig. 2). Second, we took several measures to account for possible confounds. For example, we considered only isolated errors; made sure to include an equal number of errors and position-matched sequences, which were used as a control condition; minimised the effect of transient activity bursts at block onsets; and applied both hypothesis-driven (ROI-based) and exploratory whole-brain analyses. Finally, we also conducted a replication study analyzing another data set acquired for a sub-sample of participants in a separate experimental session; the pattern of results derived from this analysis was similar to that observed in the main study

Reviewer

So, I cannot see how the data can support an argument on valence, which would seem to need separation of positive and negative prediction outcomes. There are no unexpected positive outcome.

Response

We would like to thank the reviewer for this thoughtful comment that is providing us with an opportunity to make some clarifications about the theoretical framework of our study. We did not intend to explore brain correlates underlying unexpected positive and negative outcomes, nor did we aim to test their dissociative role in valuation-related processes. Instead, building on extensive animal and human research into reward and decision signals in the brain, we assumed that the nucleus accumbens (NAc) – a primary efferent target of dopamine neurons within the mesolimbic pathway – is implicated in signaling valence. As such, it should exhibit dissociative activity patterns to unexpectedly better and worse outcomes, which are weighted relative to prospective expectations, broadcasting positive and negative valence prediction error signals (PES), respectively. However, increased dopaminergic activity within the NAc has been reported not only during unexpected rewards, which may indicate positive valence, but also during the processing of salient events, without any reward, feedback or motivational value, and during action initiation. Although these observations and the role of the NAc in signaling positive valence are not mutually exclusive, they complicate the interpretation of activity increases within this structure during unexpected positive outcomes. Thus, activity increases within the NAc to unexpectedly better outcome may indicate salience and can not be unequivocally interpreted as positive valence PES. Negative effects within the NAc during unexpected negative outcomes, on the other hand, are incompatible with the salience and action initiation account – brain circuits involved in these processes should exhibit activity increases – and, therefore, constitute a more reliable neural signature for valence PES.

Following the reviewer's comment, we added a supplementary note explaining this point. We refer to this note in the Discussion section within the following context:

“In addition to being salient events, movement errors during skilled motor routine also constitute a sudden violation of predicted success. They indicate a failure to achieve the desired outcome and, therefore, may trigger negative valence PES within the reward circuits (Hernádi et al., 2015). These negative signals can then be used to assign blame to actions that led to this failure (Wolpert et al., 2011). Our results indicate that the NAc – a structure within the ventral striatum and a valence center in the mammalian brain (O'Doherty et al., 2003; Oleson et al., 2012; Rutledge et al., 2010) (for the recent review, see O'Doherty et al., 2017) – may be involved in such retroactive negative signaling. This dopamine-innervated subcortical region did not exhibit significant changes in activity at the error onset. However, following error commission, its activity levels significantly decreased and, by the time the task performance was restored, were significantly lower than during errorless performance (Fig. 5b & 6b; see also Fig. 7 & Supplementary Fig. 8). Such disengagement of the NAc is incompatible with the salience or action initiation account (Syed et al., 2015; Zink et al., 2003) (Supplementary Notes). Instead, it may indicate a pause in dopamine transients in response to errors (Hart et al., 2014; Mirenowicz and Schultz, 1996; Oleson et al., 2012) and, to the best of our knowledge, constitutes the first fMRI evidence of negative valence PES elicited by movement errors during self-guided skilled motor routine” (page20-21, lines 375-386).

In the current study, we used the brain signature of negative valence to address the question of how prediction error signals during movement errors are coded in the brain. Thus, our study was conducted using a hypothesis-driven

REVIEWER 1

approach. We revised the relevant parts in the Introduction section making sure that our approach is comprehensively explained.

Specifically, after summarizing previous studies on neural correlates of salience and valence processing, we introduce the knowledge gap and our research questions as follows:

“Despite the theoretical assertion of salience and valence PES in the brain, it is unknown whether and how error processing during skilled motor performance relies on neural substrates underlying their computations” (page 4, lines 59-60).

“...we aimed to elucidate (1) whether error processing during skilled motor performance is associated with changes in brain regions implemented in computations of salience and valence PES, and if so, (2) how these changes evolve throughout different phases of error processing” (page 5, lines 77-79).

Operationally, we addressed these questions by studying errors that were spontaneously committed by participants when they were engaged in a motor sequence task. The advantages of such an approach are also explained in the Introduction section:

“Participants were asked to memorize a short 5-element sequence and to tap it repeatedly on a keypad using their non-dominant (left) hand (Albouy et al., 2015) (Fig. 1a). The sequence was reproduced in a continuous and self-paced manner without relying on any external cue or input. These conditions closely resemble situations in everyday life when errors constitute deviations from the sequential skilled motor movements pre-planned in advance. It is in contrast to errors committed during speeded reaction-time paradigms (Ullsperger et al., 2014). Within such a framework, the unpredicted order of stimuli requires exogenous attention and virtually eliminates the possibility to plan motor responses in advance. In addition, in the current study, no feedback nor incentive, in any form, were given to participants. The feedback-free mode of performance not only rules out the possibility of attributing changes associated with errors to the external cue, but also implies that error detection and subsequent performance recovery would be realized via internally-driven prediction error mechanisms” (page 4, lines 62-71).

Finally, we also elaborate on why, given these task conditions, we assumed that movement errors would generate negative valence PES:

“... due to highly accurate performance during the task employed in the current study (Albouy et al., 2015; Korman et al., 2003), errors would constitute unexpected events triggering unsigned salience PES. Concurrently, such sudden violation of the predicted success would also evoke negative valence PES” (page 5, lines 82-84).

REVIEWER 2

We would like to thank the reviewer for his/her careful reading, detailed summary and enthusiastic acknowledgment of the significance of our work. We are pleased to know that “On the whole this study addresses an important and relevant issue in the field of cognitive/motor control, i.e., distinguishing the nature of different error signals in the brain, especially w/r/t salience and value. The approach taken to answering this question seems sensible, and the major results appear to be of sufficient interest for the readership”.

We have taken the reviewer’s remarks as an opportunity to improve the quality and readability of our work. Please, see below our point-by-point response to each comment.

Major comments

Reviewer

Some aspects of the experiment design are unclear or ambiguous. For example, I guess the term 'matching sequence' means instances when the subject correctly reproduced the learned sequence of key presses (?), but it is not clearly defined anywhere in the manuscript that I could find, and it seems from text on lines 298-299 errorless performance is not the same thing as the matching sequence... this is one example of the design needs to be made more clear.

Response

We would like to thank the reviewer for pointing out this ambiguity. By the term “sequence”, we mean one correct trial, i.e., a set of five elements organized in the predetermined (sequential) order. To be considered as a “matching sequence”, the correct trial should meet two additional requirements: (1) it should be isolated from errors and be surrounded by at least three correct trials in a row, which all together form continuous errorless periods, i.e., error-free control condition, and (2) it should be located at the same within-block position as the error. We reconsidered this term and now use “position-matched sequence”, instead. Given that errors of interest were also surrounded by at least three correct trials in a row, so called isolated errors, we ended up with two comparable, in terms of their length, periods/conditions with and without errors. The main conclusions of the current study are based on the results obtained from the direct comparisons between these two conditions.

In the revised version of the manuscript, we now provide a clearer definition for each trial type/condition and explain why and how they were selected in the very beginning of the Results section.

“The main units of interest in our analysis are trials, i.e., sequences and errors, comprising several keypresses ...” (page 7, lines 105-106).

“Five consecutive keypresses that followed the predetermined order (i.e., sequence) were considered as one correct trial. Any other combination of consecutive keys (or a single key), including instances of incomplete sequence, between two correct trials constitutes an error” (page 7, lines 109-111).

“...only isolated errors surrounded by at least three correct trials in a row, hence forming periods with seven consecutive trials, were analyzed (Fig. 1a). If the last error within the block was followed by only two consecutive correct trials, that error was also considered” (page 7, lines 114-116).

“Control task condition. To determine the specificity of error-related changes, periods without errors were used as a control task condition. Like periods with errors, these periods comprised seven (or six if the period was at the very end of the block) consecutive trials. To account for changes in performance and in neural activity associated with fatigue (Rickard et al., 2008), time-on-task (Grinband et al., 2011), and the position of the trial within blocks (see fMRI results below), periods with and without errors were pseudo-randomly matched by their within-block position (Methods). This position was determined by the fourth trial, which during periods with and without errors constituted an error and so-called position-matched sequence, respectively. Potential differences in the statistical power between the two conditions (i.e., periods with and without errors), were minimized by including the same number of periods in each condition.” (pages 7-8, lines 124-131).

In addition, we thoroughly revised other parts of the Results and Methods sections clarifying our experimental design and better explaining our analytical approach. We also added a new panel to Fig. 1 (panel b) with examples of errors included in the analysis.

REVIEWER 2

Reviewer

The analyses that contribute to fig. 5 (A,B,C), pg. 15, seem to run afoul of one of the classical neuroimaging fallacies. Specifically, the authors claim that, using errorless performance as a baseline, 'salience' network regions are specifically engaged on error and NOT matching sequences. It appears they base this on a significant difference from the BOLD response in those regions to the errorless baseline. However, just because the BOLD signal is different from baseline in one condition, and not different from baseline in another condition, this does not indicate that the BOLD signal in condition 1 is different from the BOLD signal in condition 2. I note that they do not do this in the analyses supporting figure 3, so perhaps they could make it clear that they are not basing all their claims on the fig. 5. analyses.

Response

The reviewer is right, and we regret this confusion. Before addressing this issue, we would like to bring to the *reviewer's attention* that when revising the manuscript, we added new figures. Therefore, figure numbers in the current version of the manuscript are different from the previous version. Thus, previous Fig. 3 and 5 are now Fig. 4 and 6, respectively.

We approached the analysis of the fMRI data in the following order. First, we assessed the main effect of errors and their position-matched sequences, versus rest, separately (Fig. 4a and Fig 4c upper panels, respectively). Next, we directly contrasted between the errors and position-matched sequences to test error-specific changes (Fig. 4b). Then, we also analyzed pre- and post-error trials to assess error-related activity over time. Our main results are shown in Fig 5 and Table 3. These analyses were conducted to characterize how neural changes associated with error processing evolve over time. To do so, activity levels during the pre-error trials, errors and post-error trials were estimated against the corresponding trials of the control error-free task condition. Thus, zero contrast values (the reference line marked by the x axis) correspond to activity levels during position-matched periods of errorless performance. We also estimated changes in activity during errors and their position-matched sequences, versus rest, separately (Fig. 6). This analysis, however, was done as a post-hoc/supplementary step to visualize the differences between the two conditions and to validate their dissociation. In the figure, we include asterisk symbols to indicate significant differences not only for errors and their position-matched sequences versus rest (red and blue asterisks, respectively) but also between these two conditions (black asterisks).

We believe that given the definition of each trial type and condition – this information is currently provided in the Results section – it is now clearer that position-matched sequences are, in fact, part of the error-free condition/errorless performance. It also seems that the term “*baseline*”, which was previously used to refer to either errorless performance or rest, also introduced unclarity. Therefore, instead of using this term, we now specify what condition was used as a reference.

Following the reviewer's comment, we thoroughly revised the Results section making sure that all necessary details are provided and that the description of the fMRI data analysis is comprehensive and clear.

Reviewer

The authors note increased activity in the salience network prior to the occurrence of an error; this does not seem to fit well with their story about the 'instantaneous' recruitment of the salience network in response to errors; if the salience network responds instantly to error, why should it be active during correct performance preceding an error. I think there are ways one could support an unsigned PE story for these periods, but it doesn't appear that the authors have attempted to do so. I would like a more thorough treatment of how increased pre-error activity relates to the underlying theory supporting the authors predictions.

Response

We appreciate the Reviewer's suggestion and we acknowledge that the reviewer's remark is accurate. We did not attempt to elaborate on the idea of proactive salience PES in the Introduction section because it was not part of our hypothesis. Nevertheless, we respectfully disagree with the reviewer that the idea of the instantaneous recruitment of the salience network during error commission presented in the Introduction contradicts the suggestive interpretation of our findings in the Discussion, where we propose that salience prediction error signals (PES) may be generated proactively even before the error is actually expressed behaviorally.

In the current study, the task, during which participants generated a given sequence of movements repeatedly, was performed continuously without relying on any external cue, feedback or insensitive of any kind; the only exception was the change in the color of the fixation cross which indicated when to start and when to stop the performance. Thus, akin to real-life situations, successful task performance relied on pre-established internal representations of highly skilled motor movements that were pre-planned in advance. Therefore, errors committed by participants were neither a product of unexpected changes in the external factors nor a failure to rapidly select and generate an appropriate response to various stimuli. Instead, they derived from the internal causes, which, as we suggest, could unexpectedly increase the probability of flaws in the intended action. Our conjecture of the proactive salience PES refers to the possibility that, during skilled motor behavior, the salience network may respond to these unexpected flaws not only retroactively after the actual error occurred but also proactively, even before the negative consequences of these flaws become evident behaviorally. Thus, in both cases, the salience network generates PES in response to an unexpected deviation from the pre-planned sequential action. Whereas the retroactive mechanism relies on the somatosensory input, the generation of the salience PES proactively does not rely on such feedback but rather depends on the reliability of internal representations of actual outcomes.

Although we did not consider the idea of the proactive salience PES in our initial hypothesis, when providing an overview of evidence suggesting that the dACC is involved in signaling salience PES, we do mention that “... it has been suggested that the dACC may generate PES in situations with increased error likelihood (Alexander and Brown, 2011; Brown and Braver, 2005). It may accommodate proactive mechanisms of error detection during highly skilled piano performance (Ruiz et al., 2009)” (page 3, lines 46-48). During the revision, we slightly rephrased this statement to make a clearer distinction between these two complementary views. The former view refers to the PES reflecting error likelihood based on the estimation of situational factors, such as task difficulty, stimulus features, etc., regardless of rapidly formed internal representations of the forthcoming action or its actual outcome. The latter view, on the other hand, refers to the alerting signals that are a product of the detected flaws in internal representations of the forthcoming action. We elaborate on this distinction later in the Discussion (please, see below).

The reason for not expanding our research hypothesis of the salience PES to the possibility of their role in anticipating and potentially preventing actual errors is twofold. First, we hypothesized that error commission would be preceded by increased speed, in line with the speed-accuracy trade-off account. We introduced this hypothesis as follows:

“At the behavioral level, we hypothesized that slowing in performance would constitute a behavioral signature of error detection (Rabbitt and Vyas, 1970). We also expected that error instances would be characterized by speed-accuracy trade-off (Fitts, 1954) – a phenomenon that has been consistently reported during speeded reaction-time tasks (Laming, 1968). Such a trade-off suggests that faster performance increases the probability of errors and, therefore, should precede error commission. It also entails that reinstatement of accurate performance after errors would compromise the speed leading to post-error slowing” (page 4, lines 72-76).

Such speeding up could possibly indicate greater reliance on automatic processes and decreased cognitive control. It is also possible, however, that faster performance could be a product of greater attention. In that case, given the design of the current study, it would be impossible to disentangle between these attentional processes and proactive PES.

Second, the low temporal resolution of the fMRI signal limits the timing precision and does not allow us to reliably determine the exact moment of neural changes. As we explain in the Method section: “Due to summation effects across consecutive closely-spaced (in time) trials (Friston et al., 1998), the temporal accuracy of neural changes associated with error processing is biased by the adjacent trials. Therefore, we expected that onset of error-specific changes in BOLD signal would be shifted in time. In addition, the sampling time of 2.65 seconds used to acquire fMRI data, which is longer than an average duration of a single sequence, imposed additional limitations on timing precision so that changes associated with error commission could be actually captured in the BOLD signal of the preceding trial, as well as endure over a few consecutive trials. Given this limitation, the inferences about differential contribution of particular networks to error processing were made based on shape and peak/dip latency of temporal activity patterns determined by contrast values of relative changes immediately before, during and immediately after errors” (page 28, lines 569-577).

REVIEWER 2

Thus, without behavioral evidence for early onset of slowing, which we consider as a behavioral signature for detection of actual errors and, possibly, of flaws in intended action, the assumption of proactive salience prediction error signaling is not justified.

*Our behavioral results indicate (1) no speed-accuracy trade-off during correct trials preceding errors and (2) significant slowing during errors with its early onset even before the first wrong key was pressed. The distinction between the first error key and the first wrong key within the error is now explained in the manuscript and visualized schematically in Fig. 1b. As we mentioned before, an error was a group of all consecutive keys, between two correct trials, that did not follow the predetermined order of the sequence. Accordingly, it could comprise not only wrong keys that violated the order of the sequence but also instances of partially-completed sequence. **“Thus, the first wrong keypress that violated the predetermined order of the sequence could occur after up to four keys pressed correctly.”** (page 7, lines 111-112).*

The early onset of slowing in performance associated with errors and early activity increases within the salience network suggest that flaws in the desired action can be detected by the brain even before the consequences of these flaws become evident behaviorally. Therefore, following the reviewer’s comment, we now expose the reader to this idea already in the Results section as follows:

“This pattern of results suggests that the salience network, namely the dACC and aIns/IFC, responds instantaneously to flaws in the ongoing (or intended) action, hence, broadcasting salience PES” (page 14, lines 248-250).

Now we also include additional clarifications in the relevant paragraph of the Discussion section:

“It has been proposed that the dACC may generate salience PES not only retroactively in response to the unpredicted outcome, but also proactively in situations with increased error likelihood (Alexander and Brown, 2011; Brown and Braver, 2005). This view suggests that the dACC tracks the probability of errors based on the estimation of situational factors, such as task difficulty, stimulus features, etc. When the error likelihood is increased, the proactive PES in the dACC may drive the brain towards desirable outcome by inhibiting inappropriate actions (Ridderinkhof, 2002) and/or selectively activating more appropriate ones. It is also possible, however, that the dACC may generate proactive alerting signals in situations with fixed external conditions when the probability of errors unexpectedly increases due to internal causes. For example, during continuous motor action well-defined in advance and carried out in a stimulus-free mode, the salience network may respond to the unexpected flaws in the intended action even before the negative consequences of these flaws become evident behaviorally and the actual error occurs. Although with caution because of the limitations imposed by the low temporal resolution of the fMRI signal, we speculate that in the current study, the activity peak within the salience network shifted toward the pre-error phase (Fig. 5a & 7a) could potentially indicate such proactive salience PES. These alerting signals may forecast error commission during skilled motor performance enacting inhibitory control even before the erroneous movement is fully executed (Ruiz et al., 2009). Indeed, our results indicate that slowing in performance occurred as early as the error onset, even before the first wrong key was pressed (Fig. 3b & 3c), and was also associated with rapid recruitment of the pre-SMA and wide-spread suppression of the sensorimotor circuits, including bilateral primary sensorimotor cortices and putamen. Previous human and animal studies suggest that the pre-SMA is crucially involved in outright suppression of the initiated action (Aron and Poldrack, 2006; Garavan et al., 2003; Isoda and Hikosaka, 2007; Sharp et al., 2010; Swann et al., 2012). This medial prefrontal region is a node within the fast STN-mediated hyperdirect stopping mechanism that has global (i.e., not specific to the currently engaged task representations) suppressive effects on the motor system (Aron and Verbruggen, 2008; Greenhouse et al., 2012; Majid et al., 2012). Therefore, it is plausible that slowing in performance at the error onset observed in the current study reflects an unsuccessful attempt of the brain to prevent the forthcoming error by imposing global motor suppression. The efficiency of this effort may depend on the capacity of the salience network to anticipate forthcoming error and generate alerting PES in advance, before the erroneous movement is actually carried out” (page 20, lines 351-374).

Reviewer

Given the authors' explicit aim to dissociate value PEs from salience PEs, especially in the realm of control, it seems appropriate to acknowledge recent work that likewise attempts to dissociate value and surprise signals in control,

REVIEWER 2

specifically Vassena et al., 2020, Nature Human Behavior. Full disclosure: I am an author on that paper, but it legitimately seems relevant to this study.

Response

We would like to thank the reviewer for bringing to our attention his/her recently published paper. Acknowledging the relevance and importance of this work, we now refer to it in the Discussion section as follows:

“Seeking for a unified theoretical account, it has been proposed that the dACC plays a central role in detecting salience by estimating discrepancy between the predicted and actual outcomes (Alexander and Brown, 2011; Vassena et al., 2020)” (page 19, lines 339-341).

Reviewer

From your behavioral analyses, it appears that there is some amount of response slowing in the PreE1 periods; dACC activity is frequently correlated with time on task (e.g., Grinband et al., 2010). It appears that the fMRI analyses do not incorporate parametric modulators that might be used to model response time or sequence time. It could be possible that the increased salience network activity observed in preE periods might be related to increases in RT. They authors should probably re-rerun the analyses with RT included as a modulator to ensure that their data could not be explained as deriving from longer sequence durations.

Response

First, we would like to make some clarifications about our behavioral results. Indeed, we did observe slight slowing in performance during the pre-error trial of $0.73 \pm 1.76\%$ (mean \pm s.e.m.). Yet, it was not significantly different than zero ($t(48) = -0.22, p = 0.824$), hence indicating no significant change in performance speed immediately before the error. Thus, rather than suggesting pre-error slowing, this result violated our expectation of speeding up as a precursor of error commission. To make it clearer, we rewrote the relevant part of the Results section as follows:

“We also expected to find evidence for the speed-accuracy trade-off immediately before and after errors in the form of faster and slower tapping speed, respectively. However, a significant deviation from the mean transition duration of correct trials is observed only during the post-error ($t(48) = 3.62, p = 0.001$), but not during the pre-error performance ($t(48) = -0.22, p = 0.824$), hence providing evidence for the speed-accuracy trade-off only after errors” (page 10, lines 162-166).

Second, we would like to thank the reviewer for mentioning the work by Grinband and colleagues (2011). In this work, the researchers have shown that activity within the dACC may be positively correlated with time spent on stimulus processing, so called time on task. That is particularly true when participants are engaged in speeded reaction time tasks and need to produce a specific action as quickly as possible in response to external stimuli. Within such a framework, greater task difficulty, which is usually manipulated by introducing conflicting information, imposes greater demand on the attentional and cognitive control networks, including the dACC. Behaviorally, such increase in task difficulty is associated with longer response times and higher error rate. However, in the current study, participants were engaged in a simple, self-guided motor task reproducing a short sequence of highly-skilled movements continuously, without relying on any external cue or feedback. We assumed that in such conditions, the dACC would not be heavily involved in the task maintenance and would exhibit dissociative activity patterns from those exhibited by the task-related network. We explain the rationale for this assumption in the Introduction section as follows:

“... brain regions previously identified as implicated in PES and those forming the motor-task-related network (Doyon et al., 2018) would exhibit dissociative dynamics during errors. Such dissociation would minimize the possibility that error-related changes within the salience network are driven by processes related to task maintenance – an inherent problem in studies using cognitive and attention-demanding tasks (Grinband et al., 2011)” (page 5, lines 84-88).

The current findings suggest that, indeed, the dACC and motor-related regions exhibit dissociative neural dynamics during error processing (Fig. 4-7, Supplementary Fig. 8) and are in line with our assumption.

Finally, we would also like to clarify that in the current study, “The sequence was reproduced in a continuous and self-paced manner without relying on any external cue or input” (page 4, lines 63-64). The “GO” and “STOP” cues,

REVIEWER 2

indicating respectively the beginning and the end of each block, were the only external cues presented to participants during the experimental session (Fig. 1a). In such performance mode, time on task is not determined by the duration of discrete sequence trials but rather by the time passed from the beginning of the performance block.

Following the reviewer's comment, in addition to validating that periods with and without errors – the latter were used as a control condition – have the same within-block position, we also extracted time on task until the onset of each error and position-matched trial from the control condition (i.e., position-matched sequence) and showed that there was no difference:

“The time spent on task from the beginning of the block until the trial onset is also comparable between the two conditions ($t(48) = 1.63, p = 0.110$)” (page 8, lines 137-138).

Reviewer

This task doesn't appear to carry an explicit reward manipulation, i.e., subjects are not given any incentive (monetary or otherwise) to perform the sequences correctly. I don't necessarily think that such an incentive is absolutely required, but given the long history of studying reward/valence PEs with explicit rewards, the authors should spend some time justifying why the signals they observe in, e.g., NAc are in fact reflecting reward PEs and not some other signal.

Response

Indeed, in the current study, there was no external feedback nor incentive in any form given to participants. In the Introduction section, we added the following statement to make this point explicit and clear:

“In addition, in the current study, no feedback nor incentive, in any form, were given to participants” (page 4, lines 68-69).

Regarding the role of the NAc in reward and valence PES, it was one of our main assumptions. Building on extensive animal and human research into reward and decision signals in the brain, we assumed that the NAc is implicated in signaling valence. As such, it should exhibit dissociative activity patterns to unexpectedly better and worse outcomes, which are weighted relative to prospective expectations, broadcasting positive and negative valence PES, respectively. We introduce this idea in the Introduction section as follows:

“The neural correlates of valence PES have been initially demonstrated in animal models using reinforcement learning, during which improved performance maximizes the received reward (Schultz, 2015). It has been posited that dopamine neurons in the midbrain ventral tegmental area (VTA) may be implicated in generation of valence PES. These neurons fire in response to unpredicted reward and pause in response to unexpected omission of reward. Similarly, signed reward-related PES have also been detected in the human brain using functional magnetic resonance imaging (fMRI). Modulation of neural activity associated with reward has indeed been identified in the dopamine target areas including the ventral striatum, particularly in the nucleus accumbens (NAc) (O'Doherty et al., 2003; Rutledge et al., 2010) – a primary efferent target of VTA neurons within the mesolimbic pathway (Haber and Knutson, 2010)” (page 4, lines 49-56).

In the discussion section, we summarize our findings and re-introduce the idea of putative involvement of the NAc in generating reward and valence PES suggested by previous studies as follows:

“In addition to being salient events, movement errors during skilled motor routine also constitute a sudden violation of predicted success. They indicate a failure to achieve the desired outcome and, therefore, may trigger negative valence PES within the reward circuits (Hernádi et al., 2015). These negative signals can then be used to assign blame to actions that led to this failure (Wolpert et al., 2011). Our results indicate that the NAc – a structure within the ventral striatum and a valence center in the mammalian brain (O'Doherty et al., 2003; Oleson et al., 2012; Rutledge et al., 2010) (for the recent review, see O'Doherty et al., 2017) – may be involved in such retroactive negative signaling. This dopamine-innervated subcortical region did not exhibit significant changes in activity at the error onset. However, following error commission, its activity levels significantly decreased and, by the time the task performance was restored, were significantly lower than during errorless performance (Fig. 5b & 6b; see also Fig. 7 & Supplementary Fig. 8)” (page 20-21, lines 375-383).

REVIEWER 2

Following the reviewer's comment, we now make an explicit statement that the "... disengagement of the NAc is incompatible with the salience or action initiation account (Syed et al., 2015; Zink et al., 2003) (Supplementary Notes)" (page 21, lines 383-384). We also added the following Supplementary Note with additional explanation:

"A neural signature of valence PES. There is evidence suggesting that the NAc may be involved in processing of salient events without any reward, feedback or motivational value (Downar et al., 2003; Münte et al., 2008; Zink et al., 2003) (for meta-analysis, see Bartra et al., 2013; Fouragnan et al., 2018; Garrison et al., 2013; Wilson et al., 2018) and may also promote action initiation (Syed et al., 2015). Although these observations and the role of the NAc in signaling positive valence PES are not mutually exclusive, they complicate the interpretation of activity increases within this structure during unexpected positive outcomes. Negative effects within the NAc during unexpected negative outcomes, on the other hand, are incompatible with the salience and action initiation account – brain circuits involved in these processes should exhibit activity increases – and, therefore, constitute a more reliable neural signature for valence PES".

We continue our discussion in the main text referring to previous animal and human studies that used rewards, aversive stimuli and negative feedback to investigate how negative outcome affects behavior and dopaminergic activity:

"The disengagement of the NAc was rather prolonged and did not vanish by the second post-error sequence. This observation is consistent with lingering reduction in NAc dopamine concentration in response to aversive stimuli in animals (Roitman et al., 2008). Dips in dopamine levels are thought to affect synaptic plasticity in the indirect striatal pathway (Shen et al., 2008). Traditionally, this pathway has been considered as a "NoGo" route that suppresses actions (Albin et al., 1989; DeLong, 1990) in a behaviorally-specific, context-dependent manner (Mink, 1996). It is presumably implicated in aversive memory in mice (Hikida et al., 2010; Kravitz et al., 2012), disfavor of unrewarded targets in monkeys (Hong and Hikosaka, 2011; Nakamura and Hikosaka, 2006) and learning from negative outcomes in humans (Frank et al., 2004; Jocham et al., 2014). Accordingly, the lingering disengagement of the NAc observed in the current study may indicate enduring learning-from-error signals that impose suppression of neural populations representing the erroneous action. The selective "punishment" of undesirable representations during correct performance immediately after the error may refine selective activation of neural populations that can reliably produce the target sequence of movements, thereby promoting formation of a dedicated pathway for the effortless and stereotyped execution of the practiced motor skill (Gabitov et al., 2015, 2019b). Thus, negative PES elicited retroactively in response to errors have the potential to facilitate learning. This idea is supported by previous studies showing that, in humans, punishment-based monetary feedback leads to significantly faster online motor responses (Wächter et al., 2009) and accelerates error-based motor learning (Galea et al., 2015).

Previous evidence also indicates that motor learning can benefit from negative feedback even without subsequent monetary loss, but only if it is directly linked to actual errors in performance (Galea et al., 2015). This finding resonates with the conjecture that negative valence PES may facilitate learning via selective suppression targeting neural populations that represent the erroneous action. If error-related disengagement of the NAc reported here mediates such selective suppression, it should not have negative effect on performance once it has been recovered. Indeed, lingering disengagement of the NAc following errors did not prevent participants from rapidly accelerating their tapping speed to the levels similar to errorless performance (Fig. 3c). Thus, rather than imposing a negative impact, neural processes underlying such disengagement may, instead, facilitate post-error performance recovery, and possibly learning, by enacting prolonged selective suppression of neural populations that provoked the error" (pages 21-22, lines 387-409).

Minor comment

Reviewer

the word 'acclaim' does not mean what you think it means... I think you want the word 'claim' in those cases.

Response

We would like to thank the reviewer for pointing out this lexical mistake. We replaced it by the word "assertion".

REVIEWER 3

We are very pleased with the reviewer's remark that "The task is interesting, the N is impressive, and the data is fairly compelling".

We would like to thank the reviewer for taking the time and effort necessary to review the manuscript. We sincerely appreciate all valuable comments and suggestions, which we believe helped us to improve substantially the readability of the manuscript.

Reviewer

1. The paper is difficult to understand. I have to take responsibility for some of that – it is likely that part of my difficulty is merely due to me not being smart enough, or well-versed enough in some of the subject matter. It is also partly due to the fact that English may not be the first language of the authors. Here and there are some tell-tale mistakes in word choice. For example, page 4, line 67, "upon such conditions"; page 17, line 336, "theoretical acclaim showing that movement..."; and page 18 line 363, "caution is imposed because of the low temporal resolution..." These are small language errors that I will not dwell on. I do suggest the authors have a native English speaker proofread the work.

One identifiable, general problem in the writing is that the authors often use overly long sentences, trying to cram too many pieces of information into one point. For example, the first two sentences on page 12 (lines 235-243):

"Next, we assessed temporal characteristics of error-specific changes in activity by estimating changes in BOLD responses immediately before, during and immediately after errors relative to those evoked during continuous periods of errorless performance immediately before, during and after matching sequences, respectively."

And then

"Thus, we conducted both region of interest (ROI) and whole-brain analyses applying repeated measures ANOVA approach on individual contrast values and activation maps, respectively, obtained from pairwise comparisons between estimated BOLD responses during (1) penultimate sequences before errors and before matching sequences, (2) last sequences immediately before errors and before matching sequences, (3) errors and matching sequences, (4) first sequences immediately after errors and after matching sequences, and (5) second sequences after errors and after matching sequences."

I understand that in the later sentence, lists tend to make long sentences, but I still argue that both sentences are made harder to follow by being too long. I would advise/request that the authors go through the paper carefully with the goal of breaking the longest sentences into two or more shorter, more succinct sentences. I realize that some of the points the authors are trying to make are complex and nuanced, but still, breaking such points into smaller sections with a clear progression of what needs to be understood in what order could greatly improve the readability of this work.

I also have to note that most of my points have to do with not understanding the description, so in each case, re-writing portions of the text may be the solution.

Response

Following the reviewer's comment, we thoroughly reviewed the manuscript and reconsidered our word choices. Below are several examples.

- (1) "*upon such conditions ...*" was replaced by "*Under certain conditions ...*" (page 4, line 56) or "*These conditions closely resemble situations ...*" (page 4, line 64).
- (2) "*... theoretical acclaim ...*" was replaced by "*... theoretical assertion...*" (page 4, line 59; page 19, line 325)
- (3) "*... caution is imposed because of the low temporal resolution ...*" was rephrased as "*Although with caution because of the limitations imposed by the low temporal resolution ...*" (page 20, lines 360-361)
- (4) "*...an internally-guided motor sequence ...*" and "*... internally-guided skilled motor routine(s) ...*" was replaced by "*... a self-guided motor sequence task ...*" (page 4, line 62) and "*... self-guided skilled motor routine(s) ...*" (page 19, line 336; page 21, line 386), respectively.

REVIEWER 3

We also made substantial efforts to improve the readability of the manuscript breaking down long sentences into smaller and more accessible pieces of information. Some examples are listed in the table below.

Previous version	New revised version	Page Lines
Analysis of distribution of these matched trials within- and between blocks showed no significant results, indicating that the pseudo-random matching procedure resulted in equivalent distribution between errors and matching sequences within and across performance blocks.	We conducted a detailed analysis to test for heterogeneity of trial distribution within and across blocks. None of the results are statistically significant ..., suggesting non-heterogeneous and comparable distribution of periods included in each condition (i.e., periods with and without errors).	8 132-137
By separating performance periods from the interleaving periods of rest using a boxcar function (i.e., block design), and by modeling trials of different types (i.e., errors and sequences of interest) as events, such approach allowed us to assess blood-oxygen-level-dependent (BOLD) signal changes associated with specific trials during ongoing task performance, while controlling for mean task-related activity.	Performance periods were modeled as blocks/epochs, whereas trials of interest (i.e., errors and sequences) were modeled as events. To minimize the effects of various trial durations, trial-related changes in the blood-oxygen-level-dependent (BOLD) signal were estimated using a stick function with zero duration (Methods).	12 200-203
Next, we assessed temporal characteristics of error-specific changes in activity by estimating changes in BOLD responses immediately before, during and immediately after errors relative to those evoked during continuous periods of errorless performance immediately before, during and after matching sequences, respectively.	To characterize temporal dynamics in neural activity associated with error processing, we assessed changes across trials during the periods with errors relative to those without errors...	14 241-242
The analysis of temporal activity patterns within the SMC showed that, in both hemispheres, activity levels progressively decreased before the error, reached the minimum at the error onset and were gradually restored during the subsequent performance recovery with no evidence for lateralization.	Regions within the task-related network, on the other hand, exhibit changes in the opposite direction. Specifically, activity within the SMC – a region that exhibits lateralized activation during the task – reaches its minimum at the error onset and is gradually restored during the subsequent performance recovery with no evidence for lateralization.	14 261-264

Reviewer

2. Exactly what point in an error was the fMRI time-locked to? It seems that when the participant got a sequence wrong at any point, that the entire sequence as labelled an error. Does that mean that the participant could get item 1 through 4 right, press the wrong key for item 5, and then the onset of the error is still locked to the onset of the first (correct) key press?

Response

The reviewer is correct. Indeed, the main units of interest in our analysis were trials (i.e., sequences and errors) comprising several keypresses. Specifically, an error comprised all wrong keypresses that violated the predetermined order (i.e., sequence) and/or incomplete sequences with missing keys. In that way, the first wrong keypress within the error was not necessarily the very first error key but could also occur after up to four keys pressed in the correct sequential order. In other words, the very first error key could be either a wrong key or a correct key composing an instance of the partially completed sequence. In the analysis of the fMRI data, a stick function, which was used to estimate the effect of errors, was time-locked to the first error key.

In the revised version of the manuscript, we integrated the following changes to better explain our approach. First, we added an additional panel to Fig. 1 (panel b) showing schematically the distinction between the first error key and the first wrong key within the error. Second, we also added an explicit explanation to our approach and its rationale in the first paragraph of the Results section:

“The main units of interest in our analysis are trials, i.e., sequences and errors, comprising several keypresses (Fig. 1a). Such an approach suits better the low temporal resolution of the fMRI signal allowing us to assess neural dynamics across trials in a space-resolved manner. To get insights into the types of errors and changes in performance during errors, behavioral data were also analyzed at the level of single keypresses.

Five consecutive keypresses that followed the predetermined order (i.e., sequence) are considered as one correct trial. Any other combination of consecutive keys (or a single key), including instances of incomplete sequence, between two correct trials constitutes an error (Fig. 1b). Thus, the first wrong keypress that violated the predetermined order of the sequence could occur after up to four keys pressed correctly” (page 7, lines 105-112).

Finally, we performed an additional whole-brain analysis shifting the error onsets to the first wrong keypress and showed that the activation patterns derived from this analysis almost perfectly overlap with those derived from our original analysis of error-related effects (Supplementary Fig. 4).

Reviewer

3. Related to determining errors, what happened if the participant made a single keypress error on the first item, and then instead of going to what would normally be the second item, repeated the first item correctly and then continued on. Would that make all following key presses an error? That is, how did the authors handle errors that were merely cases where the sequence was offset by one?

Response

We would like to address this reviewer’s comment using an example. The target sequence in our study was 4-1-3-2-4. The specific case that the reviewer is referring to, as we understand it, can be visualized as follows:

... 4-1-3-2-4- ~~4~~-4-1-3-2-4 ...

... 4-1-3-2-4-**Ek**-4-1-3-2-4 ... (**Ek** stands for any single keypress)

As we explained above, any combination of keypresses that did not result in a correct trial of five consecutive keys that formed the sequence (i.e., 4-1-3-2-4), between two correct trials, were considered as an error. Therefore, in that case the error would comprise only one keypress. If this key was 4, then the error did not contain any wrong key but instead constituted an incomplete sequence with only one key pressed correctly. Otherwise, the first and only error key would be the wrong key.

During our revision, we performed an additional analysis of errors at the level of single keypresses to get insights into the types of errors considered in the analysis. The distribution of errors by their length (i.e., the number of keys) and types is shown in Fig. 2 – a newly-added figure in the manuscript. Out of 203 errors considered in the analysis, only 20 errors were one-key long. Only six out of these 20 errors were so-called incomplete sequences.

Reviewer

4. Page 8 line 170: “we further reaffirmed that slowing associated with error commission occurred as early as at the error onset, persisted during the error and reached its peak at the error offset”

REVIEWER 3

Is “at error onset” here the transition from the first erroneous press to the next press, or the transition from the last correct press to the first incorrect press?

Response

A transition that we are referring to as an “error onset” is a time interval between the last keypress of the correct trial immediately before the error to the first keypress of the error.

In the revised version of the manuscript, we added the following explanation to clarify this point:

“To estimate speed during correct trials, the mean of four within-sequence transitions was calculated, whereas during errors, transitions to the first and from the last error key were also considered (Fig. 1a) – below, these transitions are referred to as error onset and offset, respectively” (page 10, lines 153-155).

Reviewer

5. The “matching sequences” were matched according to their within-block position, meaning an error on block 1 could be matched to an errorless sequence on block 14, is that right? There appears to be an effect across blocks on transition duration. Why would the authors match to across blocks instead of just matching to periods of errorless performance within the same block?

Response

The reviewer is correct. Sequences at the same within-block position as errors were used as a control condition. Accordingly, within each pair of the position-matched trials, the sequence and the error were from different blocks. The main reason for implementing such an approach was our observation of significant differences in activity between trials within blocks. Before assessing how neural activity changes with error commission, we analyzed blocks that comprised only correct trials to characterize dynamics in the blood-oxygen-level-dependent (BOLD) signal captured by the stick function. This analysis showed robust within-block changes in the BOLD signal that were replicable and consistent across blocks. In the revised version of the manuscript, we now include detailed statistics of this analysis in the Results section of the main text as follows:

“Before analyzing error-related changes, we assessed neural signals captured by the stick function during errorless blocks (Supplementary Fig. 3). These signals fluctuate robustly following a consistent pattern within blocks with no significant differences between blocks ($F(11, 8162) = 72.79, p < 0.001$; $F(13, 742) = 1.48, p = 0.12$; $F(143, 8162) = .78, p = 0.97$, the effect of trial, the effect of block and trial by block interaction respectively). In our analysis, the effects of these within-block changes were mitigated by using a control task condition with errorless periods matched to errors based on their within-block position (see above)” (page 12, lines 204-209).

Estimating behavioral and neural correlates of errors against position-matched sequences also permitted us to account for other confounding factors previously reported in the literature, including within-block changes in performance and in neural activity associated with fatigue (Rickard et al., 2008) and time-on-task (Grinband et al., 2011). We reconsidered our explanation in this regard and now include it in the main text as follows:

“To account for changes in performance and in neural activity associated with fatigue (Rickard et al., 2008), time-on-task (Grinband et al., 2011), and the position of the trial within blocks (see fMRI results below), periods with and without errors were pseudo-randomly matched by their within-block position (Methods). This position was determined by the fourth trial, which during periods with and without errors constituted an error and so-called position-matched sequence, respectively. Potential differences in the statistical power between the two conditions (i.e., periods with and without errors), were minimized by including the same number of periods in each condition.” (pages 7-8, lines 126-131).

Before proceeding with the behavioral and fMRI data analysis, we made sure that errors and position-matched sequences are distributed in a non-heterogeneous and equivalent manner across blocks. We also estimated the time spent on task from the beginning of the block until the trial onset and showed that it was also comparable between the two conditions. We report these results in the Results section of the manuscript as follows:

“The distribution of errors and position-matched sequences within and across blocks is shown in Fig. 2d. We conducted a detailed analysis to test for heterogeneity of trial distribution within and across blocks. None of the results are statistically significant (effect of within-block position: $F(6, 318) = 1.10, p = 0.36$; effect of block: $F(13, 624) =$

REVIEWER 3

1.326, $p = 0.193$; block by trial type interaction: $F(8.95, 429.76) = 1.307, p = 0.231$), suggesting non-heterogeneous and comparable distribution of periods included in each condition (i.e., periods with and without errors). The time spent on task from the beginning of the block until the trial onset is also comparable between the two conditions ($t(48) = 1.63, p = 0.110$)” (page 8, lines 132-138).

Reviewers' comments:

Reviewer #1 (Remarks to the Author):

The careful revisions to the paper appear to have addressed (but not solved) the various issues raised in my original review.

I think the paper is acceptable, but the study has weak points and, in the limit, these are because of the study design, low temporal resolution of the fMRI methods, and the subsequent "trial" level analysis of errors. This forces analysis that does not really correspond to the actuality – for fMRI errors are assumed to be coincident with the start of the trial (and the data in Fig 2 shows in reality errors are distributed throughout); and likewise, the duration of an error trial is assumed to be zero, whereas it will be variable, depending on where in the sequence it occurs. It seems unlikely that participants would predict errors in advance of a trial (unless the error is on the first keypress). So, the "predictive" components of the fMRI results are not especially convincing for me. Finally, as I mentioned previously, the design has only spontaneous errors of negative valence, so the separation of salience and valence relies – I think – entirely on assumptions about these neural circuits, rather than by experimental manipulation that would allow their dissociation.

Reviewer #2 (Remarks to the Author):

In their revised manuscript, the authors have adequately addressed most of my major concerns with their initial manuscript. Aspects of the design and analysis approaches that were confusing or ambiguous have been clarified. I appreciate their additional discussion on the possible roles of salience PEs in behavior and NAc as signaling valence PEs.

However, I do have lingering concerns regarding the correlation of time-on-task with ACC activity I raised in my previous review. The authors argue that time-on-task effects are not relevant to this study for a couple reasons.

First, the authors provide new analysis showing that transitions within the PreE1 sequence were not significantly longer than for errorless performance (presumably using only the 4 within-sequence transitions as indicated on lines 153-154). However, there is a significant increase in the length of the transition from the PreE1 sequence to the Error sequence, and an even more pronounced increase for transitions from Error to PostE1 (Fig 3c).

Due to the design of the study, which involves continuous, self-paced performance, it is ambiguous how to model these transition periods. The authors decided to model the onset of a trial as the first key-press of the sequence, and it doesn't appear that their fMRI analyses include a regressor specifically modeling the between-sequence transitions (e.g., a stick function at the time a sequence is completed). That is, in the fMRI analyses presented, the transition from one sequence to the next is included as part of the first sequence.

Judging from average transition duration (Fig 2A) and depending on the block, it requires 1200ms (later blocks) to 2000ms (earlier blocks) to complete a sequence (i.e., 4 within-sequence transitions from the 1st button in the sequence to the 5th), well within the TR of scanner (2650ms). In other words, effects that occur following the completion of a sequence and prior to the onset of the next could be driving BOLD activity in PreE1 - increases in sequence time ought to include the between-sequence transition that is modeled as being part of a sequence in your fMRI analyses.

Furthermore, although I focused on PreE1 in my previous comments, the time-on-task issue also pertains to the Error sequence (where longer within-sequence transitions are observed, as well as

increased between-sequence transitions from Error to PostE1), and I suggested including RT modulators generally rather than specifically for the PreE1 period.

The authors additionally argue that time-on-task effects are mainly relevant when a task involves processing external cues and feedback, particularly when control demands are manipulated, and that the simple sequence task, which does not involve external cues, does not significantly involve active task maintenance. This view seems at odds with some current theory (e.g., Shenhav et al., 2013) suggesting that control networks could monitor any number of relevant signals, including those which are internally generated (for example: progress through, completion of, or deviations from a sequence), in order to support task performance.

If time-on-task is genuinely irrelevant to the sequence task in the manuscript, including sequence time (four within-sequence transitions as well as the between-sequence transition) as a parametric modulator should have minimal effect on the analyses. However, this analysis seems necessary in order to rule out this alternative explanation for the authors' findings.

Reviewer #3 (Remarks to the Author):

The authors have satisfactorily addressed my comments.

REVIEWER #1

REVIEWER #1

We would like to express our satisfaction *with the reviewer's comment that* "The careful revisions to the paper appear to have addressed ... the various issues raised in my (*reviewer's*) original review." We are also glad to read that the reviewer thinks "...that the paper is acceptable".

The reviewer also mentions, however, that "...the study has weak points and, in the limit, these are because of the study design, low temporal resolution of the fMRI methods, and the subsequent "trial" level analysis of errors."

Whereas some of these limitations cannot be solved in our current work, we believe that it does not diminish the significance of our findings. Below are our point-by-point responses to the remaining concerns brought up by the reviewer.

Reviewer

This forces analysis that does not really correspond to the actuality – for fMRI errors are assumed to be coincident with the start of the trial (and the data in Fig 2 shows in reality errors are distributed throughout); and likewise, the duration of an error trial is assumed to be zero, whereas it will be variable, depending on where in the sequence it occurs.

Response

The issue of relatively low temporal resolution of the fMRI signal is present in all neuroimaging studies using this cutting-edge technology. In the context of the current study, this limitation did not allow us to estimate changes in neural activity at the level of single keypresses. However, it was not our objective and we did not seek to characterize error processing on a scale of milliseconds, but rather sought to determine what brain mechanisms underpin computations of prediction error signals during skilled motor behavior in humans – a question that, to the best of our knowledge, has not been directly addressed before. The acquisition of the whole-brain fMRI data and the subsequent set of analyses allowed us to capture changes in activity across the entire brain that occur on a slower timescale in a space-resolved manner.

Although the analysis of the imaging data was carried out at the trial level, it is sufficiently sensitive and precise to capture specific neural changes associated with spontaneous deviations from the expected behavior (i.e., errors). During our previous revision, we conducted additional analysis and showed that activity levels at the first error key, which may correspond to correctly initiated sequence, and at the first wrong key are almost identical not only across the entire group but also for each individual (Supplementary Fig. 4).

In the current revision, we also estimated the effect of tapping speed on the magnitude of error-related changes in brain activity in two ways: (1) by estimating trial-related changes in the BOLD signal using brief epoch-related design with trial duration, and (2) by directly assessing the effect of tapping speed using parameter modulators with transition durations within and between trials (separate models for each trial phase). We did not find any significant changes in activity associated with trial durations or tapping speed during periods with errors. The results of these analyses are now included in Supplementary Materials as Supplementary Fig. 6 & 7, respectively, and are mentioned in the main text as follows:

REVIEWER #1

“Results using statistical models with actual trial duration (Supplementary Fig. 6) are consistent with the ones reported below” (page 14, lines 243-244).

“We found no evidence that activity levels within these regions are significantly modulated by time spent on each error phase (Supplementary Fig. 7)” (page 14, lines 248-249).

Reviewer

It seems unlikely that participants would predict errors in advance of a trial (unless the error is on the first keypress). So, the “predictive” components of the fMRI results are not especially convincing for me.

Response

The predictive components of the fMRI results are suggested only as a possibility when we interpret our findings in the Discussion section (page 20, lines 351-374). As the reviewer repeatedly mentioned before, the low temporal resolution of the fMRI signal limits the timing precision and does not allow us to reliably determine the exact moment of neural changes. Therefore, we draw up our conjecture based not only on the early onset of the error-specific changes in the fMRI signal but also, and primarily, on significant slowing in performance that was evident already at the error onset, even before the first wrong key was actually pressed (Fig. 3b & 3c). Further studies are needed to provide more convincing support to this conjecture.

Reviewer

Finally, as I mentioned previously, the design has only spontaneous errors of negative valence, so the separation of salience and valence relies – I think – entirely on assumptions about these neural circuits, rather than by experimental manipulation that would allow their dissociation.

Response

The reviewer is correct. In addition to our previous response to the very same issue brought up by the reviewer in his previous review, we would like to reiterate in the current reply that we did not aim at resolving some controversies related to possibly overlapping functions of mesocortical and mesolimbic networks and their dissociative role in processing salience and reward/valence. Our primary goal was to determine what brain mechanisms underpin computations of prediction error signals during skilled motor behavior in humans – a question that, to the best of our knowledge, has not been directly addressed before. To do so, we first specified cortical and subcortical networks that exhibit significant changes during action errors (Fig. 4), and then conducted a more detailed trial-by-trial analysis to characterize the time course of these changes. Building upon accumulated knowledge about the functional role of the key regions within the salience, reward and task-related networks, here we provide evidence for dissociative contribution of dopamine systems in the brain – namely mesocortical, mesolimbic and nigrostriatal pathways with the dACC, nucleus accumbens and sensorimotor striatum as their afferent targets – to execution and adaptation processes during ongoing skilled motor behavior.

REVIEWER #2

REVIEWER #2

We are pleased to read that in our revised manuscript, we "...have adequately addressed most of my (*reviewer's*) major concerns..." and that "...aspects of the design and analysis approaches that were confusing or ambiguous have been clarified".

Yet, the reviewer does "...have lingering concerns regarding the correlation of time-on-task with ACC". We addressed these concerns by conducting additional analyses. Please, see our point-by-point responses *to the reviewer's comments* below.

Reviewer

First, the authors provide new analysis showing that transitions within the PreE1 sequence were not significantly longer than for errorless performance (presumably using only the 4 within-sequence transitions as indicated on lines 153-154). However, there is a significant increase in the length of the transition from the PreE1 sequence to the Error sequence, and an even more pronounced increase for transitions from Error to PostE1 (Fig 3c).

Due to the design of the study, which involves continuous, self-paced performance, it is ambiguous how to model these transition periods. The authors decided to model the onset of a trial as the first key-press of the sequence, and it doesn't appear that their fMRI analyses include a regressor specifically modeling the between-sequence transitions (e.g., a stick function at the time a sequence is completed). That is, in the fMRI analyses presented, the transition from one sequence to the next is included as part of the first sequence.

Judging from average transition duration (Fig 2A) and depending on the block, it requires 1200ms (later blocks) to 2000ms (earlier blocks) to complete a sequence (i.e., 4 within-sequence transitions from the 1st button in the sequence to the 5th), well within the TR of scanner (2650ms). In other words, effects that occur following the completion of a sequence and prior to the onset of the next could be driving BOLD activity in PreE1 - increases in sequence time ought to include the between-sequence transition that is modeled as being part of a sequence in your fMRI analyses.

Response

The reviewer is correct. Activity levels associated with each trial type (i.e., correct trials immediately before and after errors, errors, and trials during position-matched periods of continuous errorless performance) were estimated using event-related design (i.e., zero duration) with onsets set to the first keypresses of these trials. To preserve the variance explained by each covariate, BOLD signal changes associated with adjacent trials were estimated using separate models. The continuous nature of the task and the sampling resolution of the fMRI data (one data point every 2.65 seconds), which is longer than the average duration of a single trial, did not allow us to isolate BOLD changes associated with a single keypress/transition or with a single trial. Thus, despite modeling events of interest with zero durations, changes in the BOLD signal estimated for a given trial could also capture, to a small extent, some of the effects associated with between-trial transitions and with adjacent trials. We acknowledge this limitation in the first paragraph of the Results section (page 7, lines 105-107) and in the Discussion (page 20, lines 360-361). We also elaborate on our analytic approach in the Methods section (pages 26-28).

REVIEWER #2

Nevertheless, *following the reviewer's comment*, we conducted additional analyses to estimate changes in the BOLD signal time-locked to the last keypress of the trial. In the detailed information presented below we refer to this time-point as the offset of this trial, which in turn is also the onset of the subsequent trial. Parameter estimates for each trial type at its first and last keypress, versus rest, and contrast values between periods with and without errors (i.e., error and control condition, respectively) are shown in the figure below.

If we understand correctly, the reviewer's concern is that increased activity within the dACC may be associated with a longer time to process and press the keys, either in the wrong or correct sequential order, rather than with the salience PES triggered by a failure to generate the sequence. However, the direct comparison between the periods with and without errors shows that activity levels within the dACC are already significantly higher at the onset of the correct trial preceding the error ($t(48) = 3.35, p = 0.002$) (the upper panel in the figure below). The magnitude of this change, versus the control condition, is comparable to that of the error onset ($t(48) = 0.17, p = 0.865$), although the error onset itself and the subsequent transitions within the error are significantly longer than transitions within and between correct trials (Fig. 3b & 3c). The longest transition with the greatest slowing was observed at the error offset – the transition from the last error key to the first key of the first post-error sequence (Fig. 3b & 3c). However, by the error offset, the dACC activity drops down dramatically and does not differ significantly from the control condition ($t(48) = 0.19, p = 0.848$). Thus, these results provide no evidence that increased activity within the dACC is directly linked to longer tapping time, not only during the trials themselves, but also between trials. These findings are, however, in line with the role of the dACC in generation of the salience PES as we hypothesized in the Introduction section. Note that the direction and the time-scale of changes in activity within the dACC are dissociable from changes within the nucleus accumbens (NAc) (the middle panel in the figure below). Finally, during periods with errors, both the dACC and NAc show changes in the opposite direction from the ones exhibited by the primary sensorimotor regions (the lower panel in the figure below).

Figure. Temporal characteristics of changes in activity during error processing in the key regions of the salience, valence/reward-related and task-related networks (upper, middle and lower panel, respectively; left and right plots show changes in the left and right hemisphere, respectively). Activity levels during periods with and without errors versus rest (red and blue lines, respectively) and their contrast values (dark gray lines; values along the x-axis correspond to periods without errors, i.e., the control condition) were extracted from models with event-related design (i.e., zero durations) with onsets at the first (highlighted data-points) and the last trial keys. The first error-key and the first wrong key within the error (marked with the red oval) were compared to the first key of the position-matched sequence. The ROI within the dorsal anterior cingulate cortex (dACC) overlaid on the mean structural image of all participants is also shown. Black asterisks indicate significant differences in activity between corresponding trials and phases during periods with and without errors. */**/*** – significant results at 0.05/0.01/0.001 level; in the latter case, the results are significant after Bonferroni correction for 25 ROIs included in the analysis ($p < 0.002$).

Reviewer

Furthermore, although I focused on PreE1 in my previous comments, the time-on-task issue also pertains to the Error sequence (where longer within-sequence transitions are observed, as well as increased between-sequence transitions from Error to PostE1), and I suggested including RT modulators generally rather than specifically for the PreE1 period.

Response

We addressed the reviewer's concern about the "time-on-task" issues related to slowing in performance associated with error commission in two ways: (1) by estimating trial-related changes in the BOLD signal using a brief epoch-related design modeling the actual trial duration, and (2) by directly assessing the effect of tapping speed using parameter modulators with transition durations within and between trials (separate models for each trial phase). We did not find any significant activity increases within the dACC, or other brain regions, associated with longer transition durations during periods with errors. The results of these analyses are now included in Supplementary Materials as Supplementary Fig. 6 and Supplementary Fig. 7, respectively, and are mentioned in the main text as follows:

"Results using statistical models with actual trial duration (Supplementary Fig. 6) are consistent with the ones reported below" (page 14, lines 243-244).

"We found no evidence that activity levels within these regions are significantly modulated by time spent on each error phase (Supplementary Fig. 7)" (page 14, lines 248-249).

Reviewer

The authors additionally argue that time-on-task effects are mainly relevant when a task involves processing external cues and feedback, particularly when control demands are manipulated, and that the simple sequence task, which does not involve external cues, does not significantly involve active task maintenance. This view seems at odds with some current theory (e.g., Shenhav et al., 2013) suggesting that control networks could monitor any number of relevant signals, including those which are internally generated (for example: progress through, completion of, or deviations from a sequence), in order to support task performance.

Response

We agree with the view suggesting the existence of internal performance monitoring mechanisms and do not argue against the role of control networks in monitoring continuous self-guided and stimulus-free performance. Our hypothesis about the role of the dACC in salience PES during skilled motor action does not contradict the conflict monitoring model. According to this model, monitoring system may silently run on the background seeking for deviations from the expected outcome.

However, we do argue that the networks involved in the active task maintenance are specific to the requirements and the type of the task at hand. In the current study, participants were engaged in continuous production of highly skilled finger movements in a predetermined sequential order.

REVIEWER #2

Previous imaging studies, conducted by our and other research groups, constantly showed that successful performance in this type of task primarily relies on sensorimotor cortical and subcortical circuits (e.g., Debas et al., 2010; Orban et al., 2010; Wiestler and Diedrichsen, 2013; Albouy et al., 2015; Gabitov et al., 2015, 2016, 2019; Cohen and D'Esposito, 2016; Yokoi et al., 2018; Pinsard et al., 2019). Thus, as opposed to conflict-inducing reaction time tasks, the maintenance of correct performance during the motor sequence task is supported by “automatic” processing with minimal demands from costly attentional and cognitive control circuits (James, 1890). However, in the case of occasional errors, these circuits are rapidly engaged to catch and correct deviations from the intended behavior. Optimization of such dynamic interplay between automatic and controlled behavior is presumably achieved through performance monitoring system with the dACC as its central node (Gehring et al., 1993; Botvinick et al., 2001).

Thus, we did not claim that task maintenance is irrelevant to the production of self-guided skilled movements, but rather that, under such conditions, successful task performance does not rely heavily on attentional systems, including the dACC. In support of this viewpoint, our results suggest that, whereas the dACC is not significantly activated during errorless periods (Fig. 6), this region, together with other closely affiliated functional areas, is rapidly engaged once participants fail to maintain the correct performance of the task and commit an error. The salience account further suggests that the dACC supports task performance not by being actively involved in task maintenance or in post-error performance recovery per se. Instead, the dACC role is to “inform” the rest of the brain about situations that may require increased attention and rapid adjustments in the ongoing behavior.

In the manuscript, we use the term “time-on-task” to refer to time passage between block initiation and error commission. The potential differences in activity levels due to this “time-on-task” intervals, changes in activity during periods with errors were assessed against control condition comprising errorless periods at the same within-block positions as errors.

As mentioned above, we now understand that by expressing his concern about the “time-on-task”, the reviewer may have in mind the possibility that activity increases within the dACC are related to a longer processing time to generate the keypresses themselves, rather than to salience PES. We conducted additional analyses and found no evidence for a link between activity levels within the dACC and longer transition durations.

Reviewer

If time-on-task is genuinely irrelevant to the sequence task in the manuscript, including sequence time (four within-sequence transitions as well as the between-sequence transition) as a parametric modulator should have minimal effect on the analyses. However, this analysis seems necessary in order to rule out this alternative explanation for the authors' findings.

To satisfy the reviewer's request, we generated statistical models including transition durations as parametric modulators. These parametric predictors capture the variance related to transition durations that are not captured by the trial regressors. Note that the trial regressors were modeled with the stick function (i.e., zero duration) time-locked to the first trial keys or, in a case of between-trial transitions, to the last trial keys. However, the inclusion of parametric modulators

REVIEWER #2

in the model does not control for the potential effect of various transition durations on the changes in the BOLD signal captured by the trial regressors (Büchel et al., 1998).

Grinband and colleagues (2008) have shown that the variable epoch model with actual trial durations is a more physiologically plausible representation of changes in activity associated with response time. Therefore, we conducted an additional analysis estimating trial-related changes in the BOLD signal using brief epoch-related design with trial duration. The results derived from both parametric modulation and variable epoch models are now included in Supplementary Materials as Supplementary Fig. 6 and Supplementary Fig. 7.

REVIEWER #3

REVIEWER #3

We are pleased to read that we "...have satisfactorily addressed my (*reviewer's*) comments."

REVIEWERS' COMMENTS:

Reviewer #2 (Remarks to the Author):

I thank the authors for addressing my concerns. I have nothing more to add.